# META-LEARNING REPRESENTATIONS FOR LEARNING FROM MULTIPLE ANNOTATORS

## ABSTRACT

We propose a meta-learning method for learning from multiple noisy annotators. In many applications such as crowdsourcing services, labels for supervised learning are given by multiple annotators. Since the annotators have different skills or biases, given labels can be noisy. To learn accurate classifiers from such data, existing methods require many noisy annotated data. However, sufficient data might be unavailable in practice. To overcome the lack of data, the proposed method uses labeled data obtained in different but related tasks. The proposed method embeds each example in tasks to a latent space by using a neural network and constructs a probabilistic model for learning a task-specific classifier while estimating annotators' abilities on the latent space. This neural network is meta-learned to improve the expected test classification performance when the classifier is adapted to a given small amount of annotated data. This classifier adaptation is performed by maximizing the posterior probability via the expectation-maximization (EM) algorithm. Since each step in the EM algorithm is easily computed as a closed-form and is differentiable, the proposed method can efficiently backpropagate the loss through the EM algorithm to meta-learn the neural network. We demonstrate the effectiveness of the proposed method with real-world datasets with synthetic noise and real-world crowdsourcing datasets.

## 1 INTRODUCTION

Supervised learning requires labeled data for learning classifiers. In real-word applications, the labels are often given from multiple annotators. For example, in crowdsourcing services, we can obtain the labels by outsourcing annotation tasks to multiple workers (Zhang et al., 2016; Sheng & Zhang, 2019). In medical care or cyber security, multiple experts often perform annotation because annotation is quite time-consuming and difficult (Mimori et al., 2021; Salem et al., 2021).

In such situations, an essential challenge is that the obtained labels are generally *noisy*. This is because the performance of different annotators can vary widely. For example, in a crowdsourcing service, some workers are simply spammers that provide random labels to easily earn money (Kazai et al., 2011). In medical care or cyber security, even experts have much variability in annotation performance (Raykar et al., 2010; Mimori et al., 2021). When such noisy labels are used, standard supervised learning methods cannot perform well (Song et al., 2022). Thus, many methods have been proposed for learning classifiers from noisy labeled data obtained from multiple annotators without knowing ground truth labels (Raykar et al., 2010; Kajino et al., 2012; Rodrigues & Pereira, 2018; Chu et al., 2021). These methods usually require a large amount of annotated data to deal with noisy labels. However, such data might be difficult to collect in some applications. For example, since the crowdsourcing services usually charge a fee on the basis of the number of data and annotators, enough annotated data are difficult to collect when budgets are constrained. In medical care or cyber security, due to the scarcity of data and expertise requirements, it is difficult to collect both sufficient data to be labeled and experts who can annotate.

Even if a large amount of data is difficult to collect on a task of interest, called a target task [1], sufficient data with ground truth labels might be available in different but related tasks, called

---

[1]Although "task" is often used in the sense of "example" to be annotated in crowdsourcing literature, our paper uses the "task" for a set of examples.

source tasks. For example, if we want to build medical imaging diagnosis systems in a target task, in which only a limited number of noisy annotated training images are available, we can use clean labeled data in other image classification problems (e.g., ImageNet (Deng et al., 2009).) [2]

In this paper, we propose a meta-learning method for learning classifiers from a limited number of noisy labeled data from multiple annotators on target tasks by using clean labeled data in source tasks. Figure 1 illustrates the overview of our problem setting. Meta-learning aims to learn how to learn from a few data and has recently been successfully used for various small data problems such as few-shot classification (Hospedales et al., 2020). Meta-learning is usually formulated as a bi-level optimization problem. In the inner problem, task-specific parameters of the model are adapted to a given small amount of task-specific examples in a source task. In the outer problem, common parameters shared across all tasks are meta-learned to improve the expected test performance when the adapted model is used.

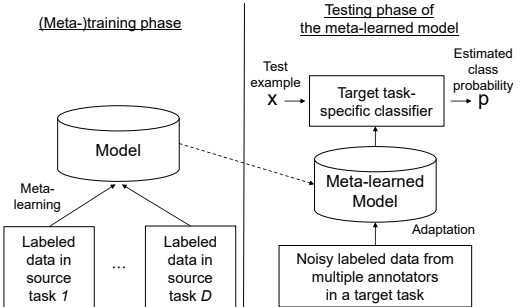

Figure 1: Our problem setting. In the (meta-)training phase, our model is meta-learned with multiple source tasks. In the testing phase of the meta-learned model, a classifier is obtained using the meta-learned model adapted to noisy labeled data in a target task.

In the inner problem of the proposed method, we first embed the given task-specific examples to a latent space by a neural network and build a probabilistic latent variable model for learning classifiers from multiple annotators on the latent space. By using the high expressive capability of neural networks, we aim to meta-learn embeddings suitable for our problem. Since source tasks contain only clean labeled data, we artificially introduce label noises to the task-specific examples at every meta-training iteration to simulate the test environment. We will demonstrate that this pseudo-annotation strategy is critical to improve performance in our experiments. With our probabilistic model, a ground truth label for each given example is treated as a latent variable, and the embedded task-specific examples are modeled by the Gaussian mixture model (GMM) on the latent space. Each component of GMM corresponds to a ground truth class label (latent variable). Further, each annotator's ability is modeled as an annotator-specific confusion matrix, whose entries are the probability of the observed noisy label given a ground truth label. This modeling allows the GMM to be fitted to the embedded data while accounting for each annotator quality. The GMM parameters and annotator-specific confusion matrices are task-specific parameters and are obtained by maximizing the posterior probability given the pseudo-annotated task-specific data and the priors with the expectation-maximization (EM) algorithm. By using the adapted GMM parameters, the task-specific classifier is obtained as the posterior probability of the ground truth label given an example in a principled manner.

In the outer problem, we meta-learn the parameters of the neural network such that the expected test classification performance, which is directly calculated with clean labeled data, is improved when the adapted classifier is used. Since each step in the EM algorithm is easily obtained as a closed-form and is differentiable, we can efficiently solve the bi-level optimization with a standard gradient descent method. By meta-learning the neural network on various tasks, we can obtain example representations suitable for learning from multiple noisy annotators on unseen tasks.

## 2 RELATED WORK

Many methods have been proposed for learning from multiple noisy annotators (Sheng & Zhang, 2019; Zhang et al., 2022). Early methods focused on estimating ground truth labels for given training data (Dawid & Skene, 1979; Whitehill et al., 2009; Venanzi et al., 2014; Welinder et al., 2010). Majority voting (MV) is the simplest method. However, it often does not work well because it ig-

---

[2]For convenience, throughout this paper, we assume that all labels in the source tasks are correct. However, it may be difficult to collect entirely clean data in real-world applications. Thus, in practice, we use datasets that are expected to have relatively low levels of noise as the source tasks. We discuss how to apply our method while explicitly treating noisy labeled data in source tasks in Section J.

nores per-annotator characteristics (Raykar et al., 2010). The Dawid and Skene (DS) model (Dawid & Skene, 1979) is a famous probabilistic model that explicitly models the annotator's ability as a class-dependent and input example-independent confusion matrix. It estimates the ground truth labels via the EM algorithm. Due to its effectiveness, the DS model has been the basis for subsequent studies (Whitehill et al., 2009; Venanzi et al., 2014; Welinder et al., 2010; Li & Yu, 2014). Later studies focused on directly learning classifiers from noisy labeled data given by multiple annotators (Raykar et al., 2010; Kajino et al., 2012; Rodrigues et al., 2013). Among them, neural network-based methods have gained attention for their high expressive capability (Albarqouni et al., 2016; Rodrigues & Pereira, 2018; Chu et al., 2021; Chu & Wang, 2021; Wei et al., 2022; Li et al., 2020; Gao et al., 2022; Guo et al., 2023; Li et al., 2024). Although these methods are promising, when there are insufficient data, they do not work well since overfitting easily occurs. Although our method also uses neural networks, they are shared across all tasks and are learned with all data in source tasks. Therefore, our method can learn from a few data without overfitting.

Meta-learning aims to learn how to learn from a few data by using data in related tasks (Hospedales et al., 2020; Vanschoren, 2018). Although most works focus on few-shot classification with clean labeled data, a few methods use the meta-leaning for learning from multiple noisy annotators (Zhang et al., 2023; Xu & Zhang, 2022; Han et al., 2021b;a). Zhang et al. (2023) and Han et al. (2021a) use a model pre-trained by ordinary meta-learning or transfer learning for estimating ground truth labels of only given examples in target tasks. Unlike the proposed method, these methods cannot directly learn classifiers on target tasks. One method requires additional annotators for testing examples in target tasks (Han et al., 2021b). Most importantly, all these methods use pre-trained models without noisy annotations as feature extractors and simply apply learning methods from noisy annotators to only target tasks. In contrast, the proposed method incorporates the classifier learning from noisy annotators during the meta-learning phase to mimic the test environment. By constructing appropriate probabilistic models in the inner optimization, the proposed method can adapt to data given from noisy annotators via the closed-form EM steps, which leads to effective and fast meta-learning. We demonstrate that this learning strategy dramatically improve performance in our experiments.

In the meta-learning methods, gradient-based methods such as model-agnostic meta-learning (MAML) (Finn et al., 2017) are a representative approach, which solves the inner problems by using gradient descent methods. They require second-order derivatives of the parameters of neural networks to solve the bi-level optimization and thus impose high computation cost (Rajeswaran et al., 2019; Bertinetto et al., 2018). The proposed method can solve the inner problem via the EM algorithm, where each EM step is easily obtained as a closed-form. Thus, the proposed method is more efficient than MAML. In addition, unlike MAML, the EM algorithm does not require the step size of gradient descents to be determined, which greatly affects meta-learning performance. Another representative approach is the embedding-based methods that meta-learns neural networks for embeddings, such as prototypical network (Snell et al., 2017). It calculates the prototype of each class as the average of the embedded training data in the same class and classifies a new example on the basis of the distance between each prototype and the embedded example. The proposed method can be regarded as a natural extension of the prototypical network. Specifically, in our probabilistic model, each prototype is obtained as the weighted average of the training data, where the weights are estimated from noisy labeled data in a principled manner. A few studies formulate the inner problem as probabilistic modeling for topic modeling or clustering (Iwata, 2021a;b; Lee et al., 2020). Our method also incorporates the probabilistic model for learning from multiple annotators in the meta-learning.

## 3 PROPOSED METHOD

In Section 3.1, we formulate our problem setting. In Section 3.2, we present our model to learn a classifier from noisy labeled data given by multiple annotators. In Section 3.3, we explain our meta-learning procedure to train our model. Figure 2 shows the overview of our meta-learning procedure.

### 3.1 PROBLEM FORMULATION

In the (meta-)training phase, we are given $D$ source tasks $\mathcal{D} = \{\{(\mathbf{x}_{dn}, t_{dn})\}_{n=1}^{N_d}\}_{d=1}^{D}$, where $D$ is the number of tasks, $N_d$ is the number of data in the $d$-th task, $\mathbf{x}_{dn} \in \mathcal{X}$ is the feature vector of

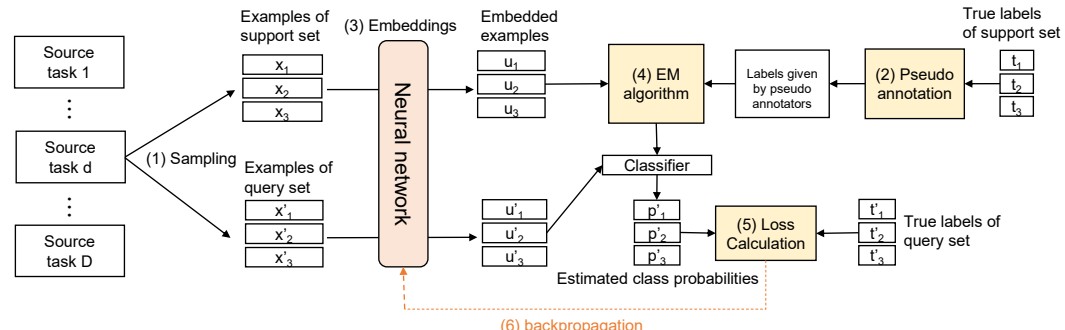

Figure 2: Overview of our meta-learning procedure. (1) For each training iteration, we randomly sample a few labeled data, called support set, and labeled data, called query set, from a randomly selected source task. (2) We generate pseudo-labels on the basis of a support set to simulate data obtained from multiple noisy annotators. (3) All examples are embedded to a latent space by a neural network, and (4) the task-specific classifier is obtained by the EM algorithm on the latent space with the support set (Section 3.2). (5) The classification loss is calculated with the classifier on the query set and (6) is backpropagated to update the neural network.

the $n$-th example of the $d$-th task, $t_{dn} \in \{c_{d1}, \dots, c_{dK_d}\}$ is its ground truth label, $c_{dk}$ is the $k$-th class of the $d$-th task, and $K_d$ is the number of classes of the $d$-th task. In the testing phase, we are given a small number of noisy labeled data from $R$ annotators $\mathcal{S} = \{(\mathbf{x}_n, \{y_n^r\}_{r \in I_n})\}_{n=1}^{N_S}$ (support set) in an unseen target task, where $y_n^r \in \{c_1, \dots, c_K\}$ is an annotation for the $n$-th example by the $r$-th annotator, $K$ is the number of classes, and $I_n \subseteq \{1, \dots, R\}$ is the index set of annotators that give labels for the $n$-th example. We assume that $I_n \neq \emptyset$, i.e., each example is assigned at least one or more labels. We make the standard assumption in meta-learning studies (Hospedales et al., 2020): the classes $\{c_1, \dots, c_K\}$ in the target task do not overlap with those in the source tasks, and feature space $\mathcal{X}$ is the same across all tasks. Our goal is to learn an accurate target task-specific classifier from noisy labeled data $\mathcal{S}$.

## 3.2 MODEL

We explain how to learn a classifier given noisy labeled data from the annotators $\mathcal{S}$. In the following, class $c_k$ is denoted as class $k$ for simplicity. We first non-linearly embed each example $\mathbf{x} \in \mathcal{X}$ to an $M$-dimensional latent space by using a neural network $f$, $\mathbf{u} = f(\mathbf{x}; \theta) \in \mathbb{R}^M$, where $\theta$ is the parameters of the neural network that is common across all tasks. We aim to meta-learn parameters $\theta$ to improve the expected test classification performance. The meta-learning procedure will be described in next section.

To obtain classifiers, we consider a probabilistic latent variable model on the latent space, where a ground truth label for the $n$-th example $t_n \in \{1, \dots, K\}$ is treated as a latent variable. We use probabilistic models because they are successfully used for learning from multiple annotators in the previous studies (Dawid & Skene, 1979; Raykar et al., 2010; Tanno et al., 2019; Kim et al., 2022). With our model, ground truth label $t$ is generated from the categorical distribution with parameters $\boldsymbol{\pi} = (\pi_k)_{k=1}^K$: $p(t = k|\boldsymbol{\pi}) = \pi_k$ where $\pi_k \geq 0$ and $\sum_{k=1}^K \pi_k = 1$. We assume that embedded example $\mathbf{u} \in \mathbb{R}^M$ is generated depending on its ground truth label $t$ as follows, $p(\mathbf{u}|t = k, \mathbf{M}) = \mathcal{N}(\mathbf{u}|\boldsymbol{\mu}_k, \mathbf{I})$, where $\mathcal{N}(\cdot|\boldsymbol{\mu}, \boldsymbol{\Sigma})$ is the Gaussian distribution with mean $\boldsymbol{\mu} \in \mathbb{R}^M$ and covariance $\boldsymbol{\Sigma} \in \mathbb{R}^{M \times M}$, $\mathbf{I}$ is $M$-dimensional identity matrix, $\boldsymbol{\mu}_k \in \mathbb{R}^M$ is a prototype for the $k$-th class, and $\mathbf{M} = (\boldsymbol{\mu}_k)_{k=1}^K$. Although we assume the isotropic variance $\mathbf{I}$ for simplicity, we can use other covariance matrices such as full covariance matrices. For the $r$-th annotator, we assume confusion matrix $\mathbf{A}_r \in [0, 1]^{K \times K}$, where its $(l, k)$-th element $\alpha_{lk}^r$ represents the probability that the $r$-th annotator returns label $l$ when ground truth label is $k$: $p(y^r = l|t = k, \mathbf{A}) = \alpha_{lk}^r$, where $\alpha_{lk}^r \geq 0$, $\sum_{l=1}^K \alpha_{lk}^r = 1$, and $\mathbf{A} = (\mathbf{A}_r)_{r=1}^R$. Such an input example-independent confusion matrix has been commonly and successfully used for modeling the annotator's ability in the previous crowdsourcing studies (Chu et al., 2021; Rodrigues & Pereira, 2018; Raykar et al., 2010; Tanno et al., 2019; Kim et al., 2022).

By using the above components, the likelihood of the embedded support examples $\mathbf{U} = \{\mathbf{u}_n\}_{n=1}^{N_{\mathcal{S}}}$ and their noisy labels $\mathbf{Y} = \{\{y_n^r\}_{r \in I_n}\}_{n=1}^{N_{\mathcal{S}}}$ is given by

$$p(\mathbf{U}, \mathbf{Y}|\mathbf{M}, \boldsymbol{\pi}, \mathbf{A}) = \prod_{n=1}^{N_{\mathcal{S}}} \sum_{t_n} \left[ p(\mathbf{u}_n|t_n, \mathbf{M}) p(t_n|\boldsymbol{\pi}) \prod_{r \in I_n} p(y_n^r|t_n, \mathbf{A}) \right]. \tag{1}$$

Note that this model is equivalent to the GMM on the latent space when excluding annotator term $\prod_{r \in I_n} p(y_n^r|t_n, \mathbf{A})$. In Eq. (1), $\mathbf{M}, \boldsymbol{\pi}$, and $\mathbf{A}$ are task-specific parameters to be estimated from support set $\mathcal{S}$. We estimate $\mathbf{M}, \boldsymbol{\pi}$, and $\mathbf{A}$ by maximizing the posterior probability as follows,

$$\mathbf{M}^*, \boldsymbol{\pi}^*, \mathbf{A}^* = \underset{\mathbf{M}, \boldsymbol{\pi}, \mathbf{A}}{\arg \max} \ln p(\mathbf{M}, \boldsymbol{\pi}, \mathbf{A}|\mathbf{U}, \mathbf{Y}) = \underset{\mathbf{M}, \boldsymbol{\pi}, \mathbf{A}}{\arg \max} \ln p(\mathbf{U}, \mathbf{Y}|\mathbf{M}, \boldsymbol{\pi}, \mathbf{A}) + \ln p(\mathbf{M}, \boldsymbol{\pi}, \mathbf{A}), \tag{2}$$

where we assume the conjugate prior distribution for each component of the likelihood [3]:

$$p(\mathbf{M}, \boldsymbol{\pi}, \mathbf{A}) = p(\boldsymbol{\pi}) \prod_{k=1}^{K} p(\boldsymbol{\mu}_k) \prod_{r=1}^{R} p(\mathbf{A}_r), \tag{3}$$

$$p(\boldsymbol{\mu}_k) = \mathcal{N}(\boldsymbol{\mu}_k|\mathbf{0}, \tau^{-1}\mathbf{I}), \ \ p(\boldsymbol{\pi}) = \frac{\Gamma(K(b+1))}{\Gamma(b+1)^K} \prod_{k=1}^{K} \pi_k^b, \ \ p(\mathbf{A}_r) = \prod_{k=1}^{K} \left[ \frac{\Gamma(K(c+1))}{\Gamma(c+1)^K} \prod_{l=1}^{K} (\alpha_{lk}^r)^c \right], \tag{4}$$

where $\tau > 0$ is a precision parameter, $p(\boldsymbol{\pi})$ is the Dirichlet distribution with the Dirichlet parameter $b > 0$, and $\Gamma(\cdot)$ is the gamma function. $p(\mathbf{A}_r)$ is the product of the Dirichlet distribution with parameter $c > 0$. By using such conjugate priors, we can stabilize the EM algorithm and derive a closed-form EM step as described later. In this paper, we treat $\tau$, $b$, and $c$ as hyperparameters.

We find a local optimum solution for Eq. (2) by using the EM algorithm, where the task-specific parameters are updated by maximizing the following lower bound $Q$ of the objective function in Eq. (2):

$$\ln p(\mathbf{U}, \mathbf{Y}|\mathbf{M}, \boldsymbol{\pi}, \mathbf{A}) + \ln p(\mathbf{M}, \boldsymbol{\pi}, \mathbf{A}) \geq$$

$$\sum_{n,k=1}^{N_{\mathcal{S}}, K} \lambda_{nk} \ln \frac{p(\mathbf{u}_n|t_n = k, \mathbf{M}) p(t_n = k|\boldsymbol{\pi}) \prod_r p(y_n^r|t_n = k, \mathbf{A})}{\lambda_{nk}} + \ln p(\mathbf{M}, \boldsymbol{\pi}, \mathbf{A}) \equiv Q, \tag{5}$$

where we used Eq. (1) and Jensen's inequality to derive $Q$, $\lambda_{nk}$ is the responsibility that represents the probability that the label of the $n$-th example is $k$-th class, $\lambda_{nk} \geq 0$, and $\sum_{k=1}^{K} \lambda_{nk} = 1$. This inequality becomes tighter as $\lambda_{nk}$ approaches true posterior $p(t_n = k|\mathbf{u}_n, \mathbf{Y}, \mathbf{M}, \boldsymbol{\pi}, \mathbf{A})$, and the equality holds if and only if $\lambda_{nk}$ matches the true posterior (Bishop, 2006). The EM algorithm alternates between the E step and M step to maximize the lower bound. Specifically, in the E step, the responsibility is calculated by analytically maximizing lower bound $Q$ w.r.t. $\lambda_{nk}$,

$$\lambda_{nk} = p(t_n = k|\mathbf{u}_n, \mathbf{Y}, \mathbf{M}, \boldsymbol{\pi}, \mathbf{A})$$

$$= \frac{p(\mathbf{u}_n|t_n = k, \mathbf{M}) p(t_n = k|\boldsymbol{\pi}) p(\mathbf{Y}|t_n = k, \mathbf{A})}{p(\mathbf{u}_n, \mathbf{Y}, \mathbf{M}, \boldsymbol{\pi}, \mathbf{A})} = \frac{\mathcal{N}(\mathbf{u}_n|\boldsymbol{\mu}_k, \mathbf{I}) \pi_k a_{nk}}{\sum_{l=1}^{K} \mathcal{N}(\mathbf{u}_n|\boldsymbol{\mu}_l, \mathbf{I}) \pi_l a_{nl}}, \tag{6}$$

where $a_{nk} := p(\mathbf{Y}|t_n = k, \mathbf{A}) = \prod_{r \in I_n} \prod_{l=1}^{K} (\alpha_{lk}^r)^{\delta(y_n^r, l)}$ and $\delta(X, Y)$ is the delta function, i.e., $\delta(X, Y) = 1$ if $X = Y$, and $\delta(X, Y) = 0$ otherwise.

In the M step, task-specific parameters $\mathbf{M}, \boldsymbol{\pi}$, and $\mathbf{A}$ are calculated as follows,

$$\boldsymbol{\mu}_k = \frac{\sum_{n=1}^{N_{\mathcal{S}}} \lambda_{nk} \mathbf{u}_n}{\tau + \sum_{n=1}^{N_{\mathcal{S}}} \lambda_{nk}}, \ \ \pi_k = \frac{\sum_{n=1}^{N_{\mathcal{S}}} \lambda_{nk} + b}{Kb + N_{\mathcal{S}}}, \ \ \alpha_{lk}^r = \frac{\sum_{n \in I^r} \lambda_{nk} \delta(y_n^r, l) + c}{\sum_{n \in I^r} \lambda_{nk} + Kc}, \tag{7}$$

---

[3] A conjugate prior is a prior distribution that produces a posterior in the same family as the prior when combined with a specific likelihood (Bishop, 2006). Since the conjugate priors make the inference simpler, they are commonly used in probabilistic modeling. In our model, since the likelihood $p(\mathbf{u}_n|t_n, \mathbf{M})$ is a Gaussian, we used the Gaussian prior (conjugate prior) $p(\boldsymbol{\mu}_k)$. Since the likelihoods $p(t_n|\boldsymbol{\pi})$ and $p(y_n^r|t_n, \mathbf{A})$ are multinomial distributions, we used the Dirichlet priors (conjugate priors) $p(\boldsymbol{\pi})$ and $p(\mathbf{A}_r)$, respectively.

which are obtained analytically by maximizing the lower bound $Q$ with respect to $\boldsymbol{\mu}_k$, $\pi_k$, and $\alpha_{lk}^r$, respectively. $I^r \subseteq \{1, \ldots, N\}$ is the index set of the examples that are labeled by the $r$-th annotator. The EM algorithm is guaranteed to monotonically increase the posterior probability at each step until it reaches a local maximum. We repeat the EM steps $J$ times for the adaptation and obtain estimated task-specific parameters $\mathbf{M}^*$, $\boldsymbol{\pi}^*$, and $\mathbf{A}^*$. As a result, the class label probability given a new example $\mathbf{u} = f(\mathbf{x})$ is calculated by

$$\ln p(t = k | \mathbf{u}; \mathcal{S}) \propto \ln p(\mathbf{u} | t = k, \mathbf{M}^*) + \ln p(t = k | \boldsymbol{\pi}^*) = -\frac{1}{2} \parallel \mathbf{u} - \boldsymbol{\mu}_k^* \parallel^2 + \ln \pi_k^*. \quad (8)$$

The class label of $\mathbf{u}$ can be predicted by $\arg\max_k[-\frac{1}{2}\|\mathbf{u} - \boldsymbol{\mu}_k^*\|^2 + \ln \pi_k^*]$. Our generative model-based formulation can naturally treat tasks with different numbers of classes. This property would be preferable in practice, e.g., when the number of classes in the target task is different from that in the source tasks. We note that when each class has the uniform prior, i.e., $\pi_k = \frac{1}{K}$, $\tau = 0$, and each example has clean label; $\lambda_{nk} = 1$ for truth class $k$, the adapted classifier in Eq. (8) is equivalent to that of the prototypical network (Snell et al., 2017), which is a well-known embedding-based meta-learning method. Therefore, our model can be regarded as a natural extension of the prototypical network for learning from multiple annotators.

### 3.3 META-TRAINING

We explain the meta-training procedure for our model. In this section, notation $\mathcal{S}$ is used as a support set with pseudo noisy labels in source tasks. In the outer optimization, meta-parameters to be optimized are parameters of the neural network $\theta$. We maximize the expected test classification performance when the classifier obtained in the inner optimization is used:

$$\min_\theta \mathbb{E}_{d \sim \{1, \ldots, D\}} \mathbb{E}_{(\bar{\mathcal{S}}, \mathcal{Q}) \sim \mathcal{D}_d} \mathbb{E}_{\{\mathbf{B}_r\} \sim p(\mathbf{B})} \mathbb{E}_{\mathcal{S} \sim p(\bar{\mathcal{S}}, \{\mathbf{B}_r\})} [-\ln p(\theta, \mathcal{Q}; \mathcal{S})], \quad (9)$$

where $\mathcal{Q} = \{(\mathbf{x}_n, t_n)\}_{n=1}^{N_\mathcal{Q}}$ and $\bar{\mathcal{S}} = \{(\mathbf{x}_m, t_m)\}_{m=1}^{N_{\bar{\mathcal{S}}}}$ are testing data, called a query set, and a support set drawn from the same source task without overlapping, respectively. Since source tasks have only clean labeled data, $\bar{\mathcal{S}}$ are also clean. Therefore, to mimic the test environment, we artificially create noisy labels of $R$ pseudo-annotator on the basis of $\bar{\mathcal{S}}$. Specifically, we first draw $R$ annotators' confusion matrices $\{\mathbf{B}_r\}_{r=1}^R$ from a predefined confusion matrix distribution $p(\mathbf{B})$, where the $(l, k)$-th element of $\mathbf{B}_r$ is $\beta_{lk}^r$ that represents the probability that the $r$-th pseudo-annotator returns label $l$ when the ground truth label is $k$. The specific form of the confusion matrix distribution will be described in Section 4.1. By using $\{\mathbf{B}_r\}_{r=1}^R$ and $\bar{\mathcal{S}}$, support set with noisy labels $\mathcal{S} = \{(\mathbf{x}_m, \{y_m^r\}_{r=1}^R)\}_{m=1}^{N_{\mathcal{S}}}$ are generated, where $y_m^r$ is a label for the $m$-the example annotated from the $r$-th pseudo-annotator and is generated by using the $r$-th confusion matrix $\mathbf{B}_r$ on truth label of the $m$-th example $t_m$. In Eq. (9), log-likelihood $\ln p(\theta, \mathcal{Q}; \mathcal{S})$ is described as

$$\ln p(\theta, \mathcal{Q}; \mathcal{S}) = \sum_{n=1}^{N_\mathcal{Q}} \ln p(t_n | \mathbf{u}_n; \mathcal{S}) =$$

$$\sum_{n=1}^{N_\mathcal{Q}} -\frac{1}{2} \parallel \mathbf{u}_n(\theta) - \boldsymbol{\mu}_{t_n}^*(\theta) \parallel^2 + \ln \pi_{t_n}^*(\theta) - \ln \left( \sum_{k=1}^K \exp(-\frac{1}{2} \parallel \mathbf{u}_n(\theta) - \boldsymbol{\mu}_k^*(\theta) \parallel^2) \pi_k^*(\theta) \right), \quad (10)$$

where we explicitly describe the dependency of neural network parameter $\theta$ for clarity. $\mathbf{M}^*(\theta)$ and $\boldsymbol{\pi}^*(\theta)$ are task-specific parameters adapted to support set $\mathcal{S}$. Since $\mathbf{M}^*(\theta)$ and $\boldsymbol{\pi}^*(\theta)$ are easily obtained by the EM algorithm, the outer problem in Eq. (9) is efficiently constructed. In addition, the outer problem is differentiable since $\mathbf{M}^*(\theta)$ and $\boldsymbol{\pi}^*(\theta)$ are differentiable w.r.t. $\theta$. Thus, we can solve it by a stochastic gradient descent method. Algorithm 1 shows the pseudocode for our meta-training procedure. For each iteration, we randomly sample task $d$ from source tasks (Line 2). From the $d$-th task, we randomly sample support set $\bar{\mathcal{S}}$ (Line 3). From $\mathcal{D}_d \setminus \bar{\mathcal{S}}$, we randomly sample query set $\mathcal{Q}$ (Line 4). We generate a support set with pseudo-noisy labels $\mathcal{S}$ from $\bar{\mathcal{S}}$ (Lines 5–6). Since source tasks contain only clean labeled data, we artificially created noisy labeled data. When source tasks have labeled data obtained from multiple annotators, we can directly use them. We initialize the responsibilities $\lambda_{nk}$ (Line 7). In our experiments, we initialized $\lambda_{nk} = \frac{1}{R} \sum_{r=1}^R \delta(y_n^r, k)$. By repeating the EM steps $J$ times, we can obtain the classifier adapted to support set $\mathcal{S}$ (Lines 8–11). Lastly, we calculate the negative log-likelihood (loss) on a query set $\mathcal{Q}$ (Line 12) and update

---

**Algorithm 1** Meta-training procedure of our model.

---

**Require:** Source tasks $\mathcal{D}$, support set size $N_\mathcal{S}$, query set size $N_\mathcal{Q}$, the number of the EM steps $J$, the number of pseudo-annotators $R$, and the confusion matrix distribution $p(\mathbf{B})$
**Ensure:** Common parameters of neural network $\theta$
1: **repeat**
2:    Randomly sample task $d$ from $\{1, \dots, D\}$
3:    Randomly sample support set $\bar{\mathcal{S}}$ with size $N_\mathcal{S}$ from $d$-th task $\mathcal{D}_d$
4:    Randomly sample query set $\mathcal{Q}$ with size $N_\mathcal{Q}$ from $\mathcal{D}_d \setminus \bar{\mathcal{S}}$
5:    Randomly sample $R$ pseudo-annotators' confusion matrices $\{\mathbf{B}_r\}_{r=1}^R$ from $p(\mathbf{B})$
6:    Generate support set with noisy labels $\mathcal{S}$ from both $\bar{\mathcal{S}}$ and $\{\mathbf{B}_r\}_{r=1}^R$
7:    Initialize the responsibilities for $\mathcal{S}$, $(\lambda_{nk})_{n=1,k=1}^{N_\mathcal{S},K}$
8:    **for** $j := 1$ to $J$ **do**
9:       Update task-specific parameters $\mathbf{M}$, $\boldsymbol{\pi}$, and $\mathbf{A}$ by Eq. (7)
10:      Update $(\lambda_{nk})_{n=1,k=1}^{N_\mathcal{S},K}$ for $\mathcal{S}$ by Eq. (6)
11:    **end for**
12:   Calculate the negative log-likelihood on $\mathcal{Q}$, $-\ln p(\theta, Q; S)$, with Eq. (10)
13:   Update common parameters $\theta$ with the gradients of the negative log-likelihood
14: **until** End condition is satisfied;

---

common parameters $\theta$ with the gradient of the loss (Line 13). After meta-learning $\theta$, given a support set from multiple annotators $\mathcal{S}$ on unseen target tasks, we can obtain the target task-specific classifier by executing Lines 7 to 11 with $\mathcal{S}$ on Algorithm 1. The time complexity of the EM algorithm (Eqs. (6) and (7)) is $\mathcal{O}(JN_\mathcal{S}K(KR + M))$. In our setting, the numbers of support data $N_\mathcal{S}$, classes $K$, and annotators $R$ are small, and we can freely control the embedding dimension $M$. Additionally, as shown in our experiments, the EM algorithm works well with small $J$. Thus, the proposed method can perform fast adaptation via the EM algorithm during the meta-learning phase.

## 4 EXPERIMENTS

### 4.1 DATA

In the main paper, we used three real-world datasets: Omniglot, Miniimagenet, and LabelMe. Omniglot and Miniimagenet are commonly used in meta-learning studies (Snell et al., 2017; Finn et al., 2017; Rajeswaran et al., 2019; Zhang et al., 2023), and LabelMe is a real-world crowdsourcing dataset that is commonly used in crowdsourcing studies (Rodrigues & Pereira, 2018; Chu et al., 2021; Chu & Wang, 2021). Omniglot consists of hand-written images of 964 characters (classes) from 50 alphabets (Lake et al., 2015). Miniimagenet consists of images of 100 classes obtained from the ILSVRC12-dataset (ImageNet) (Vinyals et al., 2016). Omniglot and Miniimagenet consist of only clean labeled data. Since meta-learning requires many tasks (classes), we used these datasets. LabelMe consists of 2,688 images of 8 classes. One-thousand of them were annotated from 59 workers in Amazon Mechanical Turk (Rodrigues et al., 2017). Each image was labeled by an average of 2.5 workers. The details of these datasets are described in Section F. We additionally evaluated the proposed method with the CIFAR-10H dataset (Peterson et al., 2019) in Section I.9.

For Omniglot, we randomly used 764 classes for meta-training, 100 classes for validation, and 100 classes for testing. For Miniimagenet, we randomly used 70 classes for meta-training, 10 classes for validation, and 20 classes for testing. For both datasets, we generated 50 tasks for both validation and testing data. For each task, we first randomly select four classes. Then, for each class of a task, we randomly used a few examples (one, three or, five examples per class) for support data and 10 examples for query data. For LabelMe, we used all eight classes for testing and created 10 tasks. We used the same number of support and query data as for Omniglot and Miniimagenet. Since LabelMe does not have many classes, we used Miniimagenet for meta-training and validation. Since LabelMe consists of classes not included in Miniimagenet and has a small number of annotators per example (2.5), this dataset is suitable for our problem setting. Thus, we used this dataset. Note that using different datasets for source and target tasks is important since such a situation would be common in practice. For each dataset, we created five different meta-training/validation/testing splits.

Since Omniglot and Miniimagenet do not have data annotated by multiple annotators, we artificially create them. Such simulated annotation is commonly used in previous studies (Han et al., 2021a; Tu et al., 2020; Tanno et al., 2019; Chu et al., 2021; Li et al., 2020; Cao et al., 2019) since it enables us to investigate the methods in controlled environments. For simulated annotations for target tasks, we considered three types of annotators: expert, hammer, and spammer. The expert returns ground truth labels with probability $q$ ($0.8 < q \leq 1$) or otherwise chooses wrong labels uniformly at random. The hammer also returns labels on the basis of the same mechanism as the experts with a low probability for returning ground truth labels $q$ ($0.5 < q \leq 0.8$). The spammer returns labels uniformly randomly from all classes. These types of annotators are typical worker types in the real crowdsourcing service and thus are commonly used in experiments of previous studies (Kazai et al., 2011; Han et al., 2021a; Tu et al., 2020). For the support set in each target task, we randomly draw $R$ annotators from a predefined annotator's distribution $p(A)$, where $A$ takes expert (E), hammer (H), or spammer (S). Specifically, we considered four cases $(p(\mathrm{E}), p(\mathrm{H}), p(\mathrm{S})) = (0.1, 0.8, 0.1), (0.1, 0.7, 0.2), (0.1, 0.6, 0.3), (0.1, 0.5, 0.4)$. After selecting the annotator type, we uniformly randomly select accuracy rate $q$ in each range when the type is expert or normal. All support data were annotated from all $R$ annotators. For our method, pseudo-annotators are generated at each meta-training iteration from only the single annotator's distribution $(p(\mathrm{E}), p(\mathrm{H}), p(\mathrm{S})) = (0.1, 0.7, 0.2)$ [4]. The confusion matrix distribution for pseudo-annotators $p(\mathbf{B})$ was constructed from the $(p(\mathrm{E}), p(\mathrm{H}), p(\mathrm{S})) = (0.1, 0.7, 0.2)$ and accuracy rate $q$.

For Omniglot and Miniimagenet, we evaluated mean test accuracy on target tasks over four different target annotator distributions mentioned above with different numbers of support data per class within $\{1, 3, 5\}$ and annotators $R$ within $\{3, 5, 7\}$. For LabelMe, we evaluated mean test accuracy on target tasks with different numbers of support data per class within $\{1, 3, 5\}$. In Section I.4, we evaluated the proposed method with other annotator's types on target tasks such as pair-wise flippers or class-wise spammers (Tanno et al., 2019; Khetan et al., 2018) when the same annotator distribution $(p(\mathrm{E}), p(\mathrm{H}), p(\mathrm{S})) = (0.1, 0.7, 0.2)$ was used for meta-learning. Even when the annotator's type is different between target and source tasks, our method worked well.

### 4.2 COMPARISON METHODS

We compared the proposed method (Ours) with 13 methods for learning from noisy annotators: two types of logistic regression (LRMV and LRDS), two types of random forest (RFMV and RFDS), two types of prototypical networks (PrMV and PrDS), two types of MAML (MaMV and MaDS), the crowd layer (Rodrigues & Pereira, 2018) (CL), a meta-learning variant of CL (MCL), a learning method from crowds with common noise adaptation layers (Chu et al., 2021) (CNAL), its meta-learning variant (MCNAL), and the proposed method without pseudo-annotators (w/o PA). We also evaluated other recent methods (Liang et al., 2022; Gao et al., 2022) in Sections I.7 and I.8.

Here, methods with the symbol 'MV' used majority voting for determining the label of each support example. Methods with the symbol 'DS' used the DS model for estimating labels of support data (Dawid & Skene, 1979). Although the DS model is simple, it has been reported to perform better than many existing methods (Zheng et al., 2017). Thus, we chose it as a comparison method. Although the original DS model does not consider the priors, we used them as in the proposed method because it improved the performance. In the DS methods, the initial value of responsibility $\lambda_{nk}$ in the EM algorithm was set to $\lambda_{nk} = \frac{1}{R} \sum_{r=1}^{R} \delta(y_n^r, k)$, where $R$ is the number of annotators in a task, as in the proposed method. When the number of the EM step $J$ is one, the initial values (soft labels) are directly used for classifier learning. CL is a famous neural network-based learning method from multiple annotators, which have annotator-specific layers. CNAL is a recently proposed neural network-based learning method from multiple annotators, which models both an annotator-invariant confusion matrix and annotator-specific confusion matrices. CL and CNAL learn the whole neural networks with gradient decent methods. LRMV, LRDS, RFMV, RFDS, CL, and CNAL used only target support data obtained from multiple annotators for learning classifiers. We included these methods to investigate the effectiveness of using data in source tasks. If these methods outperform the proposed method, there is no need to perform meta-learning in the first place. Thus, it is important to include them in comparison for extensive experiments.

---

[4] Although this simple distribution worked well in our experiments, other distributions may be more optimal. Determining a better distribution will be a future challenge.

Table 1: Average test accuracies over four target annotator distributions with different numbers of support data $N_{\mathcal{S}}$ and annotators $R$ on Omniglot and Miniimagenet. The number of classes is four, and the number of support data per class (shot) is one, three, and five. Boldface denotes the best and comparable methods according to the paired t-test ($p = 0.05$).

| $N_{\mathcal{S}}$ | $R$ | Ours | LRMV | LRDS | RFMV | RFDS | CL | CNAL | PrMV | PrDS | MaMV | MaDS | MCL | MCNAL | w/o PA |
|---|---|---|---|---|---|---|---|---|---|---|---|---|---|---|---|
| 4 | 3 | **0.692** | 0.410 | 0.422 | 0.347 | 0.358 | 0.569 | 0.425 | **0.666** | 0.687 | 0.655 | 0.678 | 0.661 | 0.511 | 0.433 |
| 4 | 5 | **0.814** | 0.456 | 0.458 | 0.382 | 0.384 | 0.641 | 0.483 | 0.769 | 0.775 | 0.758 | 0.764 | 0.754 | 0.593 | 0.458 |
| 4 | 7 | **0.855** | 0.480 | 0.485 | 0.404 | 0.410 | 0.678 | 0.485 | 0.820 | 0.834 | 0.811 | 0.823 | 0.810 | 0.606 | 0.484 |
| 12 | 3 | **0.885** | 0.498 | 0.516 | 0.438 | 0.463 | 0.700 | 0.656 | 0.825 | 0.816 | 0.698 | 0.721 | 0.778 | 0.752 | 0.794 |
| 12 | 5 | **0.938** | 0.552 | 0.568 | 0.491 | 0.511 | 0.776 | 0.719 | 0.893 | 0.891 | 0.777 | 0.799 | 0.855 | 0.819 | 0.871 |
| 12 | 7 | **0.967** | 0.608 | 0.620 | 0.539 | 0.557 | 0.836 | 0.752 | 0.943 | 0.944 | 0.847 | 0.864 | 0.912 | 0.860 | 0.924 |
| 20 | 3 | **0.930** | 0.544 | 0.562 | 0.503 | 0.535 | 0.762 | 0.732 | 0.885 | 0.871 | 0.735 | 0.760 | 0.831 | 0.805 | 0.914 |
| 20 | 5 | **0.964** | 0.606 | 0.640 | 0.566 | 0.599 | 0.837 | 0.796 | 0.936 | 0.935 | 0.817 | 0.844 | 0.900 | 0.873 | 0.959 |
| 20 | 7 | **0.982** | 0.662 | 0.688 | 0.620 | 0.644 | 0.886 | 0.826 | 0.969 | 0.971 | 0.882 | 0.900 | 0.943 | 0.903 | **0.981** |
| Avg. | | **0.892** | 0.535 | 0.551 | 0.476 | 0.496 | 0.743 | 0.653 | 0.856 | 0.858 | 0.776 | 0.795 | 0.827 | 0.747 | 0.758 |

(a) Omniglot

| $N_{\mathcal{S}}$ | $R$ | Ours | LRMV | LRDS | RFMV | RFDS | CL | CNAL | PrMV | PrDS | MaMV | MaDS | MCL | MCNAL | w/o PA |
|---|---|---|---|---|---|---|---|---|---|---|---|---|---|---|---|
| 4 | 3 | **0.387** | 0.245 | 0.248 | 0.256 | 0.258 | 0.287 | 0.276 | 0.374 | 0.380 | 0.365 | 0.367 | 0.331 | 0.294 | 0.316 |
| 4 | 5 | **0.436** | 0.246 | 0.245 | 0.258 | 0.259 | 0.293 | 0.282 | 0.405 | 0.405 | 0.394 | 0.392 | 0.353 | 0.308 | 0.349 |
| 4 | 7 | **0.432** | 0.243 | 0.243 | 0.262 | 0.261 | 0.301 | 0.280 | **0.432** | 0.429 | 0.409 | 0.407 | 0.369 | 0.305 | 0.372 |
| 12 | 3 | **0.534** | 0.286 | 0.286 | 0.272 | 0.277 | 0.355 | 0.331 | 0.443 | 0.464 | 0.425 | 0.428 | 0.426 | 0.390 | 0.403 |
| 12 | 5 | **0.571** | 0.297 | 0.298 | 0.285 | 0.288 | 0.381 | 0.349 | 0.494 | 0.510 | 0.457 | 0.467 | 0.466 | 0.425 | 0.442 |
| 12 | 7 | **0.621** | 0.304 | 0.304 | 0.292 | 0.294 | 0.398 | 0.356 | 0.540 | 0.556 | 0.498 | 0.506 | 0.496 | 0.434 | 0.490 |
| 20 | 3 | **0.595** | 0.311 | 0.312 | 0.304 | 0.312 | 0.397 | 0.369 | 0.485 | 0.516 | 0.437 | 0.454 | 0.490 | 0.451 | 0.500 |
| 20 | 5 | **0.628** | 0.320 | 0.327 | 0.319 | 0.329 | 0.426 | 0.392 | 0.553 | 0.579 | 0.490 | 0.512 | 0.535 | 0.495 | 0.561 |
| 20 | 7 | **0.674** | 0.332 | 0.333 | 0.336 | 0.340 | 0.448 | 0.403 | 0.600 | 0.616 | 0.536 | 0.553 | 0.564 | 0.508 | 0.606 |
| Avg. | | **0.542** | 0.286 | 0.288 | 0.287 | 0.291 | 0.365 | 0.338 | 0.481 | 0.495 | 0.446 | 0.454 | 0.448 | 0.401 | 0.449 |

(b) Miniimagenet

Since existing meta-learning studies for multiple annotators cannot be directly used for our setting as described in Section 2, we created various neural network-based meta-learning methods: PrMV, PrDS, MaMV, MaDS, MCL, MCNAL, and w/o PA. Specifically, as in previous studies (Zhang et al., 2023; Han et al., 2021b), they meta-learn their models with clean data in source tasks without the pseudo-annotation: they maximize the expected test classification performance, which is calculated with clean labeled data (query set), when using the model adapted to a few clean labeled data (support set) in source tasks. Then, they fine-tune the meta-learned models to target tasks with noisy support data by applying methods for multiple annotators such as MV, DS, CL, and CNAL. PrMV, PrDS, MCL, and MCNAL used the prototypical networks for meta-learning. MaMV and MaDS used MAML for meta-learning. For the proposed method, we selected the hyperparameters on the basis of mean validation accuracy. For all the comparison methods, the best test results were reported from their hyperparameter candidates. The details of network architectures and hyperparameters are described in Sections G and H.

### 4.3 RESULTS

Tables 1 show the average accuracy on target tasks with different numbers of target support data and annotators with Omniglot and Miniimagenet, respectively. We did not include the standard errors of the results due to the lack of space. The full results including the standard errors are described in Section I.12. The proposed method outperformed the other methods for all cases. As the number of annotators $R$ increased, all methods tended to improve performance. This is because ground truth labels of given support data become easy to estimate when $R$ is large. Non-meta-learning methods (LRMV, LRDS, RFMV, RFDS, CL, and CNAL) tended to perform worse than other meta-learning methods (PrMV, PrDS, MaMV, MaDS, MCL, MCNAL, and w/o PA), which indicates the effectiveness of using source tasks. The proposed method outperformed these meta-learning methods. Especially, the proposed method performed better than w/o PA by a large margin. Since the difference between both methods is whether or not the pseudo-annotation was performed during the meta-learning phase, this result shows that the pseudo-annotation is essential in our framework to learn how to learn from multiple noisy annotators.

Figure 3 shows the average test accuracies over different numbers of support data and annotators by changing the ratio of the spammers in annotators on target tasks. As the ratio of the spammer increased, the performance of all methods tended to decrease since ground truth labels became difficult to estimate. Nevertheless, the proposed method consistently performed better than other methods across all the ratios. The result suggests that the proposed method can robustly learn

Table 2: Average test accuracies with different numbers of support data $N_S$ on LabelMe. The number of classes in each task is eight, and the number of support data per class is one, three, and five. Boldface denotes the best and comparable methods according to the paired t-test ($p = 0.05$).

| $N_S$ | Ours | LRMV | LRDS | RFMV | RFDS | CL | CNAL | PrMV | PrDS | MaMV | MaDS | MCL | MCNAL | w/o PA |
|---|---|---|---|---|---|---|---|---|---|---|---|---|---|---|
| 8 | **0.414** | 0.202 | 0.208 | 0.165 | 0.173 | 0.247 | 0.240 | 0.381 | 0.375 | 0.297 | 0.287 | 0.329 | 0.314 | 0.276 |
| 24 | **0.542** | 0.261 | 0.255 | 0.243 | 0.251 | 0.359 | 0.361 | 0.514 | 0.508 | 0.404 | 0.411 | 0.509 | 0.488 | 0.412 |
| 40 | **0.605** | 0.278 | 0.271 | 0.280 | 0.276 | 0.422 | 0.426 | **0.576** | 0.571 | 0.460 | 0.464 | **0.592** | **0.593** | 0.515 |
| Avg. | **0.520** | 0.247 | 0.245 | 0.229 | 0.233 | 0.343 | 0.342 | 0.490 | 0.484 | 0.387 | 0.387 | 0.477 | 0.465 | 0.401 |

classifiers for various annotator types even when the annotator's distribution is different between target and source tasks.

Table 2 shows the average accuracy on target tasks with different numbers of target support data with LabelMe. The proposed method outperformed the other methods for all cases by effectively transferring knowledge on source tasks. This result indicates that the proposed method can improve performance on the real crowdsourcing datasets by meta-learning with other datasets, which is preferable since datasets used for source and target tasks can be different in practice.

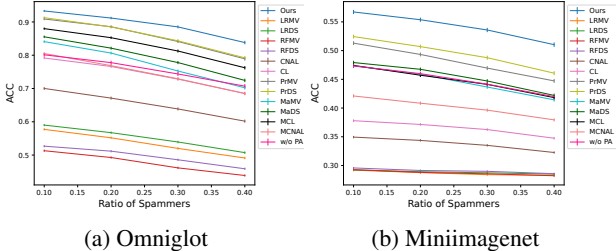

(a) Omniglot      (b) Miniimagenet

Figure 3: Average and standard errors of accuracies when changing the ratio of spammers on target tasks.

Figure 4 shows average and standard errors of test accuracies when changing the numbers of EM steps $J$. Our method consistently performed better than w/o PA over all $J$ with all datasets. Our method showed the best results when $J = 2$ or 3, which was able to select using validation data on our experiments. The result of LabelMe is shown in Section I.1.

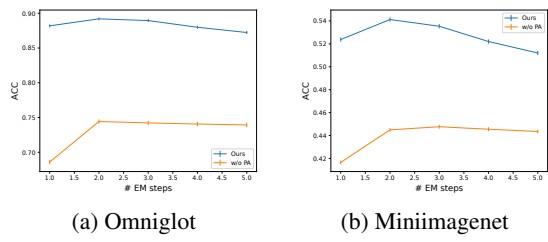

(a) Omniglot      (b) Miniimagenet

Figure 4: Average and standard errors of accuracies when changing the numbers of EM steps $J$.

We investigate the computation time for the proposed method on Omniglot. We used a Linux server with A100 GPU and 2.20Hz CPU and set the number of the EM steps to two for the proposed method, which was selected using validation data. Meta-training time of the proposed method, PrMV, and MaMV were 1361.118, 1280.697, and 3499.138 seconds, respectively. We omitted PrDS, MaDS, MCL, and MCLAL since their meta-learning processes were the same as PrMV or MaMV. Meta-testing time of the proposed method, PrMV, and MaMV were 0.960, 0.928, and 2.185 seconds, respectively. Since MaMV requires the second-order derivative of the whole parameters of the neural network for meta-learning, it took longer than other methods. The proposed method took slightly longer for meta-learning than PrMV due to the EM algorithm, but it was still fast. A more detailed discussion of the computation cost is described in Section I.11.

## 5 CONCLUSION

We proposed a meta-learning method for learning classifiers with a limited number of labeled data given from multiple annotators. Our experiments showed that our method outperformed various existing methods on real-world datasets. Although the proposed method assumes input example-independent confusion matrices for modeling annotators, it is interesting to extend our framework to handle input example-dependent confusion matrices as in (Gao et al., 2022; Guo et al., 2023; Li et al., 2024). In addition, it is also interesting to extend our framework to other crowdsourcing problems such as active learning from multiple annotators. We also plan to use variational Bayesian inference, which is also differentiable, instead of the EM algorithm in our framework.

ETHICS STATEMENTS

Since the proposed method uses data from multiple tasks for meta-learning, biased data might be included. This might result in biased results. We encourage researchers to develop methods to detect such biases automatically. In addition, although the proposed method showed good performance, there is a possibility of misclassification in practice. Therefore, this method should be used as a support tool for human decision-making. For example, in the medical image diagnosis given in Section 1, the results obtained from the proposed method are not to be used as is but as a reference (support) for a physician to make a final decision.

REPRODUCIBILITY STATEMENT

For reproducibility, we described the details of the datasets in Sections 4.1 and F, the neural network architectures in Section G, and the hyperparameters in Section H. In addition, we described the pseudo-code of our meta-learning procedure in Algorithm 1 and the detailed derivations of lower bound $Q$ of Eq. 5 and the EM algorithm (Eqs. 6 and 7) in Section C.

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

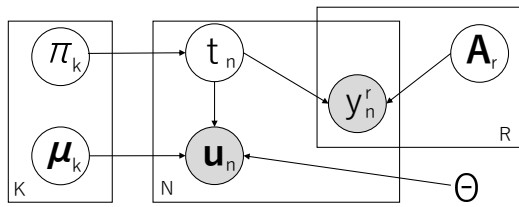

Figure 5: Graphical model representation of the proposed model in the latent space of a task. The gray and non-gray nodes represent observe and unobserved variables, respectively. Embedded example $\mathbf{u}_n$ depends on neural network parameter $\theta$ since $\mathbf{u}_n := f(\mathbf{x}_n; \theta)$.

Matteo Venanzi, John Guiver, Gabriella Kazai, Pushmeet Kohli, and Milad Shokouhi. Community-based bayesian aggregation models for crowdsourcing. In *WWW*, 2014.

Oriol Vinyals, Charles Blundell, Timothy Lillicrap, Daan Wierstra, et al. Matching networks for one shot learning. In *NeurIPS*, 2016.

Hongxin Wei, Renchunzi Xie, Lei Feng, Bo Han, and Bo An. Deep learning from multiple noisy annotators as a union. *IEEE Transactions on Neural Networks and Learning Systems*, 2022.

Peter Welinder, Steve Branson, Pietro Perona, and Serge Belongie. The multidimensional wisdom of crowds. In *NeurIPS*, 2010.

Jacob Whitehill, Ting-fan Wu, Jacob Bergsma, Javier Movellan, and Paul Ruvolo. Whose vote should count more: Optimal integration of labels from labelers of unknown expertise. In *NeurIPS*, 2009.

Sunyue Xu and Jing Zhang. Crowdsourcing with meta-knowledge transfer (student abstract). 2022.

Huaxiu Yao, Linjun Zhang, and Chelsea Finn. Meta-learning with fewer tasks through task interpolation. In *ICLR*, 2021.

Hansong Zhang, Shikun Li, Dan Zeng, Chenggang Yan, and Shiming Ge. Coupled confusion correction: learning from crowds with sparse annotations. In *AAAI*, 2024.

Jieyu Zhang, Cheng-Yu Hsieh, Yue Yu, Chao Zhang, and Alexander Ratner. A survey on programmatic weak supervision. *arXiv preprint arXiv:2202.05433*, 2022.

Jing Zhang, Xindong Wu, and Victor S Sheng. Learning from crowdsourced labeled data: a survey. *Artificial Intelligence Review*, 46(4):543–576, 2016.

Jing Zhang, Sunyue Xu, and Victor S Sheng. Crowdmeta: crowdsourcing truth inference with meta-knowledge transfer. *Pattern Recognition*, 140:109525, 2023.

Yudian Zheng, Guoliang Li, Yuanbing Li, Caihua Shan, and Reynold Cheng. Truth inference in crowdsourcing: Is the problem solved? *Proceedings of the VLDB Endowment*, 10(5):541–552, 2017.

## A  OUR GRAPHICAL MODEL

Figure 5 shows the graphical model representation of our model in the latent space of a task.

## B  EXTENDED RELATED WORK

There are some meta-learning methods for learning from a few noisy labeled data without considering multiple annotators (Liang et al., 2022; Mazumder et al., 2021; Chen et al., 2022; Killamsetty et al., 2022). Unlike the proposed method, these methods do not use information of the annotator's id for training. When the annotators' ids are available, learning methods that model multiple annotators tend to perform better than methods that do not consider annotators (Tanno et al., 2019;

Gao et al., 2022). Therefore, this paper considers modeling multiple annotators. We experimentally compared the proposed method with this approach in Section I.7.

Transfer learning is related to our work (Pan & Yang, 2009). Meta-learning is a type of fine-tuning method in transfer learning, particularly effective in very small data regimes (Hospedales et al., 2020). Since our aim is to learn from a small amount of noisy data, we focused on meta-learning. Although the standard fine-tuning methods pre-train models with clean labeled source data, we showed that it is insufficient to our setting. The proposed method performed well with pseudo-annotation during the meta-learning. Domain adaptation is a well-known transfer learning approach (Pan & Yang, 2009; Ganin & Lempitsky, 2015; Long et al., 2018). Since it simultaneously uses source and target data for training, it requires time-consuming training as new target tasks appear. In contrast, meta-learning methods do not because the meta-learned model with source data can perform rapid and efficient adaptation with only target data.

## C  DERIVATIONS OF LOWER BOUND $Q$ AND THE EM STEPS

In this section, we describe the detailed derivations of lower bound $Q$ in Eq. (5) and the EM steps in Eqs. (6) and (7).

**Lemma C.1.** $Q := \sum_{n,k=1}^{N_S,K} \lambda_{nk} \ln \frac{p(\mathbf{u}_n|t_n=k,\mathbf{M})p(t_n=k|\boldsymbol{\pi}) \prod_r p(y_n^r|t_n=k,\mathbf{A})}{\lambda_{nk}} + \ln p(\mathbf{M},\boldsymbol{\pi},\mathbf{A})$ *is a lower bound of* $\ln p(\mathbf{U},\mathbf{Y}|\mathbf{M},\boldsymbol{\pi},\mathbf{A}) + \ln p(\mathbf{M},\boldsymbol{\pi},\mathbf{A})$.

*Proof.* Since the second terms of both equations are the same, we prove that the first term of $Q$ is a lower bound of $\ln p(\mathbf{U},\mathbf{Y}|\mathbf{M},\boldsymbol{\pi},\mathbf{A})$.

$$
\begin{aligned}
\ln p(\mathbf{U},\mathbf{Y}|\mathbf{M},\boldsymbol{\pi},\mathbf{A}) &= \sum_n \ln \sum_k p(\mathbf{u}_n|t_n=k,\mathbf{M})p(t_n=k|\boldsymbol{\pi}) \prod_r p(y_n^r|t_n=k,\mathbf{A}) \\
&= \sum_n \ln \sum_k \lambda_{nk} \frac{p(\mathbf{u}_n|t_n=k,\mathbf{M})p(t_n=k|\boldsymbol{\pi}) \prod_r p(y_n^r|t_n=k,\mathbf{A})}{\lambda_{nk}} \\
&\geq \sum_{n,k} \lambda_{nk} \ln \frac{p(\mathbf{u}_n|t_n=k,\mathbf{M})p(t_n=k|\boldsymbol{\pi}) \prod_r p(y_n^r|t_n=k,\mathbf{A})}{\lambda_{nk}}, \quad (11)
\end{aligned}
$$

where we used the definition of our probabilistic model (Eq. (1)) in the first equal sign, and $\ln(\sum_k \lambda_k z_k) \geq \sum_k \lambda_k \ln(z_k)$ for any $z_k > 0$, $\sum_k \lambda_k = 1$, and $\lambda_k \geq 0$ (the Jensen's inequality) to derive the inequality. $\qquad\square$

**Lemma C.2.** *The E step of lower bound $Q$ is given by*

$$
\lambda_{nk} = \frac{\mathcal{N}(\mathbf{u}_n|\boldsymbol{\mu}_k,\mathbf{I})\pi_k a_{nk}}{\sum_{l=1}^K \mathcal{N}(\mathbf{u}_n|\boldsymbol{\mu}_l,\mathbf{I})\pi_l a_{nl}}, \quad (12)
$$

*where $a_{nk} := p(\mathbf{Y}|t_n=k,\mathbf{A}) = \prod_{r\in I_n} \prod_{l=1}^K (\alpha_{lk}^r)^{\delta(y_n^r,l)}$ and $\delta(X,Y)$ is the delta function, i.e., $\delta(X,Y) = 1$ if $X = Y$, and $\delta(X,Y) = 0$ otherwise. The M step of lower bound $Q$ is given by*

$$
\boldsymbol{\mu}_k = \frac{\sum_{n=1}^{N_S} \lambda_{nk}\mathbf{u}_n}{\tau + \sum_{n=1}^{N_S} \lambda_{nk}}, \quad \pi_k = \frac{\sum_{n=1}^{N_S} \lambda_{nk} + b}{Kb + N_S}, \quad \alpha_{lk}^r = \frac{\sum_{n\in I^r} \lambda_{nk}\delta(y_n^r,l) + c}{\sum_{n\in I^r} \lambda_{nk} + Kc}. \quad (13)
$$

*Proof.* We first derive the E step for variable $\lambda_{nk}$. Since the second term of lower bound $Q$ in Eq. (5) does not depend on $\lambda_{nk}$, we consider the first term of it (we denote this as $Q_1$). Then, we use the following general property of the lower bound (Bishop, 2006):

$$
\ln p(\mathbf{U},\mathbf{Y}|\mathbf{M},\boldsymbol{\pi},\mathbf{A}) - Q_1 = -\sum_n \sum_k \lambda_{nk} \ln \frac{p(t_n=k|\mathbf{u}_n,\mathbf{Y},\mathbf{M},\boldsymbol{\pi},\mathbf{A})}{\lambda_{nk}}. \quad (14)
$$

The r.h.s. of Eq. (14) is equivalent to the KL divergence. Since $\ln p(\mathbf{U},\mathbf{Y}|\mathbf{M},\boldsymbol{\pi},\mathbf{A})$ does not depend on $\lambda_{nk}$, maximizing $Q_1$ w.r.t. $\lambda_{nk}$ is equivalent to minimizing the KL divergence w.r.t. $\lambda_{nk}$.

Since the KL divergence takes the minimum value (zero) when two probability distributions are the same, we can maximize $Q_1$ with

$$\lambda_{nk} = p(t_n = k|\mathbf{u}_n, \mathbf{Y}, \mathbf{M}, \boldsymbol{\pi}, \mathbf{A}). \tag{15}$$

By using the Bayes' theorem, we can derive

$$\lambda_{nk} = p(t_n = k|\mathbf{u}_n, \mathbf{Y}, \mathbf{M}, \boldsymbol{\pi}, \mathbf{A})$$
$$= \frac{p(\mathbf{u}_n|t_n = k, \mathbf{M})p(t_n = k|\boldsymbol{\pi})p(\mathbf{Y}|t_n = k, \mathbf{A})}{p(\mathbf{u}_n, \mathbf{Y}, \mathbf{M}, \boldsymbol{\pi}, \mathbf{A})} = \frac{\mathcal{N}(\mathbf{u}_n|\boldsymbol{\mu}_k, \mathbf{I})\pi_k a_{nk}}{\sum_{l=1}^{K} \mathcal{N}(\mathbf{u}_n|\boldsymbol{\mu}_l, \mathbf{I})\pi_l a_{nl}}, \tag{16}$$

where $a_{nk} := p(\mathbf{Y}|t_n = k, \mathbf{A}) = \prod_{r \in I_n} \prod_{l=1}^{K} (\alpha_{lk}^r)^{\delta(y_n^r, l)}$ and $\delta(X, Y)$ is the delta function.

Next, we derive the M step for $\boldsymbol{\mu}_k$ by maximizing $Q$ w.r.t. $\boldsymbol{\mu}_k$. The derivative of $Q$ w.r.t. $\boldsymbol{\mu}_k$ is

$$\frac{\partial Q}{\partial \boldsymbol{\mu}_k} = \frac{\partial(\sum_n \lambda_{nk} \ln p(\mathbf{u}_n|t_n = k, \mathbf{M}) + \ln p(\boldsymbol{\mu}_k))}{\partial \boldsymbol{\mu}_k}$$
$$= \sum_n \lambda_{nk}(\mathbf{u}_n - \boldsymbol{\mu}_k) - \tau \boldsymbol{\mu}_k. \tag{17}$$

Thus, from the condition of $\frac{\partial Q}{\partial \boldsymbol{\mu}_k} = 0$, we can derive

$$\boldsymbol{\mu}_k = \frac{\sum_{n=1}^{N_S} \lambda_{nk} \mathbf{u}_n}{\tau + \sum_{n=1}^{N_S} \lambda_{nk}}. \tag{18}$$

Next, we derive the M step for $\pi_k$ by maximizing $Q$ w.r.t. $\pi_k$ by using the Lagrange multiplier method. Specifically, we consider Lagrange function $L(\boldsymbol{\pi}, \nu) := Q + \nu(\sum_l \pi_l - 1)$, where $\nu$ is a Lagrange multiplier. The second term of $L(\boldsymbol{\pi}, \nu)$ represents the constraint condition of $\pi_k$. The derivative of $L(\boldsymbol{\pi}, \nu)$ w.r.t. $\pi_k$ is

$$\frac{\partial L(\boldsymbol{\pi}, \nu)}{\partial \pi_k} = \frac{\partial(\sum_n \lambda_{nk} \ln p(t_n = k|\boldsymbol{\pi}) + \ln p(\boldsymbol{\pi}))}{\partial \pi_k} + \nu$$
$$= \sum_n \frac{\lambda_{nk}}{\pi_k} + \frac{b}{\pi_k} + \nu. \tag{19}$$

From the condition of $\frac{\partial L(\boldsymbol{\pi}, \nu)}{\partial \pi_k} = 0$, we can obtain

$$\pi_k = -\frac{\sum_n \lambda_{nk} + b}{\nu}. \tag{20}$$

By using the condition of $\frac{\partial L(\boldsymbol{\pi}, \nu)}{\partial \nu} = \sum_l \pi_l - 1 = 0$ and Eq. (20), we can obtain

$$\nu = -(bK + \sum_{n,l} \lambda_{nl}) = -(bK + N_S), \tag{21}$$

where we used $\sum_l \lambda_{nl} = 1$. By substituting Eq. (21) for Eq. (20), we can obtain

$$\pi_k = \frac{\sum_{n=1}^{N_S} \lambda_{nk} + b}{Kb + N_S}. \tag{22}$$

Similarly, we can derive the M step for $\alpha_{lk}^r$ by using the Lagrange multiplier method for $\alpha_{lk}^r$. Specifically, we consider Lagrange function $L(\boldsymbol{A}, \nu) := Q + \sum_{r,k} \nu_k^r(\sum_l \alpha_{lk}^r - 1)$, where $\nu := \{\nu_k^r\}_{r,k}$ are Lagrange multipliers. The derivative of $L(\boldsymbol{A}, \nu)$ w.r.t. $\alpha_{lk}^r$ is

$$\frac{\partial L(\boldsymbol{A}, \nu)}{\partial \alpha_{lk}^r} = \frac{\partial(\sum_{n \in I^r} \lambda_{nk} \ln p(y_n^r|t_n = k, \mathbf{A}) + \ln p(\boldsymbol{A}))}{\partial \alpha_{lk}^r} + \nu_k^r$$
$$= \sum_{n \in I^r} \frac{\lambda_{nk}}{\alpha_{lk}^r} \delta(y_n^r, l) + \frac{c}{\alpha_{lk}^r} + \nu_k^r. \tag{23}$$

From the condition of $\frac{\partial L(\boldsymbol{A}, \nu)}{\partial \alpha_{lk}^r} = 0$, we can obtain

$$\alpha_{lk}^r = -\frac{\sum_{n \in I^r} \lambda_{nk} \delta(y_n^r, l) + c}{\nu_k^r}. \tag{24}$$

By using the condition of $\frac{\partial L(\boldsymbol{A}, \nu)}{\partial \nu_k^r} = \sum_l \alpha_{lk}^r - 1 = 0$ and Eq. (24), we can obtain

$$\nu_k^r = -(cK + \sum_{n \in I^r} \sum_l \lambda_{nk} \delta(y_n^r, l)) = -(cK + \sum_{n \in I^r} \lambda_{nk}). \tag{25}$$

By substituting Eq. (25) for Eq. (24), we can obtain

$$\alpha_{lk}^r = \frac{\sum_{n \in I^r} \lambda_{nk} \delta(y_n^r, l) + c}{\sum_{n \in I^r} \lambda_{nk} + Kc}. \tag{26}$$

$\square$

# D    INTUITIVE EXPLANATION OF HOW EM ALGORITHMS CAN COPE WITH NOISY LABELS

This section explains how our EM algorithm described in Section 3.2 contributes to the robustness.

Suppose we know the ground truth labels of data. In that case, we can easily estimate annotators' confusion matrices (i.e., the probability that the $r$-th annotator returns label $l$ when the ground truth label is $k$; $\alpha_{lk}^r$) using the ground truth and annotators' labels. Conversely, if we know true annotators' confusion matrices, we can estimate the ground truth labels from the confusion matrices and annotators' labels. That is, ground truth labels and confusion matrices are interdependent. Thus, if we initialize the labels of data, we can alternately estimate the labels and confusion matrices. Here, we can make the label estimation more robust by considering multiple annotators' confusion matrices. If the initialization is not far from ground truth labels, we can expect to perform accurate estimation.

The EM algorithm is a systematical way to perform this procedure from a given probabilistic model. Specifically, in our setting, it iteratively estimates the label probabilities of the data ($\lambda_{nk}$) using the current estimate of annotators' confusion matrices ($\alpha_{lk}^r$) and classifier parameters ($\boldsymbol{\mu}_k$ and $\pi_k$) in Eq. (6), then uses the estimated labels to further refine the estimation of the annotator confusion matrices and classifier parameters in Eq. (7). In our experiments, simple and commonly used initialization $\lambda_{nk} = \frac{1}{R} \sum_{r=1}^{R} \delta(y_n^r, k)$, where $R$ is the number of annotators that labeled example, worked well as described in Section 3.3. This is the intuitive reason why the EM algorithm can obtain accurate parameters from multiple noisy annotators.

However, this is still a challenging estimation problem since no true labels exist. The proposed method meta-learns example embeddings that make this problem easier to solve by explicitly maximizing the performance of the estimated classifier in source tasks.

# E    DISCUSSION OF THE SITUATION WHERE THERE ARE MANY SPARSE ANNOTATORS

In some real-world applications, there are many annotators and each of them only labels a few support data. Here, we discuss this situation in two cases.

When each annotator labels 'the same' support data in a target task, accurate labels of the data can be estimated since each instance has many annotations. Thus, in such cases, the proposed method and conventional meta-learning methods that simply use support data with the estimated labels (e.g., PrMV, PrDS, MaMV, and MaDS in our experiments) would perform well. In contrast, since the number of support data is small, ordinary methods for learning from multiple annotators (crowdsourcing methods) that do not use source data cannot learn accurate classifiers even if they can estimate accurate labels of support data.

When each annotator labels (generally) 'different' support data in a target task, the number of whole support data becomes large since there are many annotators. Since we focus on a small data regime, this situation is out of the scope of our work. Ordinary crowdsourcing methods such as (Zhang et al., 2024) are more appropriate for these situations.

## F DATA DETAILS

In the main paper, we used three real-world datasets: Omniglot, Miniimagenet, and LabelMe. Omniglot and Miniimagenet are commonly used in meta-learning studies (Snell et al., 2017; Finn et al., 2017; Rajeswaran et al., 2019; Hospedales et al., 2020), and LabelMe is a real-world crowdsourcing dataset that is commonly used in crowdsourcing studies (Rodrigues & Pereira, 2018; Chu et al., 2021; Chu & Wang, 2021). Omniglot consists of hand-written images of 964 characters from 50 alphabets (Lake et al., 2015). There were 20 images in each character class, and each image was represented by gray scale with $28 \times 28$ pixels. The numbers of examples and classes were 19,280 and 964, respectively. Miniimagenet consists of images of 100 classes obtained from the ILSVRC12-dataset (ImageNet) (Vinyals et al., 2016). Each image was represented by RGB with $84 \times 84$ pixels. The numbers of examples and classes were 60,000 and 100, respectively. Omniglot and Miniimagenet consist of only clean labeled data. LabelMe consists of 2,688 images of 8 classes: "highway", "inside city", "tall building", "street", "forest", "coast", "mountain", and "open country". One-thousand of them were annotated from 59 workers in Amazon Mechanical Turk (Rodrigues et al., 2017)[5]. Each image was represented by RGB with $256 \times 256$ pixels and was labeled by an average of 2.5 workers. Since the proposed method (and most meta-learning methods) requires the same input feature space across tasks, we resized images of LabelMe to images with $84 \times 84$ pixels to match the image size between Miniimagenet and LabelMe.

## G NEURAL NETWORK ARCHITECTURES

For embedding neural networks of the proposed method, PrMV, PrDS, and w/o PA, we used the convolutional neural network (CNN) architecture that was composed of four convolutional blocks by following (Snell et al., 2017). Each block comprised a 64-filter $3 \times 3$ convolution, batch normalization layer (Ioffe & Szegedy, 2015), a ReLU activation and a $2 \times 2$ max-pooling layer. As a result, the dimensionality of embedding $M$ were 64, 1,600, and 1,600 for Omniglot, Miniimagenet, and LabelMe, respectively. MCL and MCNAL used the meta-learned model with the prototypical network for feature extraction on target tasks. For CL, CNAL, MCL, MCNAL, and MAML-based methods (MaMV and MaDS), a linear output layer was added to the CNN architecture for classifiers. All methods were implemented using Pytorch (Paszke et al., 2019). For the proposed method, we selected the hyperparameters on the basis of mean validation accuracy. For all the comparison methods, the best test results were reported from their hyperparameter candidates.

## H HYPERPARAMETERS

For LRMV and LRDS, regularization parameter $C$ was chosen from $\{10^{-4}, 10^{-3}, \dots 1\}$. For RFMV and RFDS, the number of trees was chosen from $\{10, 50, 100, 200\}$. For MaMV and MaDS, the iteration number for the inner problems was set to three, and the step size was selected from $\{10^{-2}, 10^{-1}, 1\}$. For the proposed method, LRDS, RFDS, PrDS, MaDS, w/o PA, the number of EM steps was chosen from $\{1, 2, 3, 4, 5, 10\}$. For the proposed method, we set precision parameter $\tau = 1.0$, Dirichlet parameter for confusion matrices $c = 1.0$, and selected the Dirichlet parameter for class-priors $b$ from $\{1, 10, 100\}$. For CL, CNAL, MCL and MCNAL, the number of fine-tuning iterations on target tasks was chosen from $\{10, 100, 300\}$. For CNAL and MCNAL, the regularization parameter $\lambda$ and embedded dimension for the probability of the annotator-invariant confusion matrix being chosen were set to $10^{-5}$ and 20, respectively, as in the original paper. For all methods except for the proposed method, the best test results were reported from their hyperparameter candidates. For the proposed method, we selected the hyperparameters on the basis of mean validation accuracy. For all neural network-based methods, we used the Adam optimizer (Kingma & Ba, 2014) with a learning rate of $10^{-3}$. The mean validation accuracy was also used for early stopping to

---

[5]http://fprodrigues.com/publications/deep-crowds/

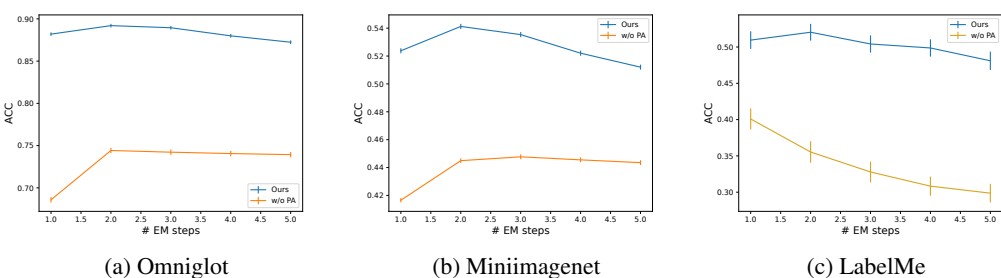

(a) Omniglot        (b) Miniimagenet        (c) LabelMe

Figure 6: Average and standard errors of accuracies with the proposed method when changing the numbers of EM steps $J$.

avoid overfitting, where the maximum number of meta-training iterations were $30,000$ and $60,000$ for Omniglot and Miniimagenet, respectively. We used a Linux server with A100 GPU and 2.20Hz CPU.

# I   ADDITIONAL EXPERIMENTAL RESULTS

## I.1   IMPACT OF NUMBERS OF EM STEPS $J$

Figure 6 shows average and standard errors of test accuracies when changing the numbers of EM steps $J$ on all datasets. The proposed method consistently performed better than w/o PA, which is the proposed method without pseudo-annotators, over all $J$ with all datasets. The proposed method showed the best results when $J = 2$ or $3$ on all datasets, which was able to select using validation data on our experiments. We note that since LabelMe has fewer target tasks for testing than other datasets, the standard errors in LabelMe were larger than the others.

## I.2   VISUALIZATION OF DATA REPRESENTATIONS

Figure 7 shows the two-dimensional visualization of the embedded data from the proposed method on Omniglot. We used three support examples per class with five annotators. Although w/o PA, which is the proposed method without pseudo-annotation, did not accurately estimate the prototype for a class (green) due to the harmful effect of noisy support data, the proposed method was able to accurately estimate the prototype of each class.

## I.3   ESTIMATED ANNOTATOR'S CONFUSION MATRICES

Figure 8 shows the estimated confusion matrices for annotators on a target task by the proposed method on Omniglot. The proposed method successfully estimated the confusion matrix for each annotator from a small amount of support data (three samples per class) by using knowledge on meta-training tasks.

## I.4   EXPERIMENTS WITH OTHER ANNOTATOR'S TYPES

We evaluated the proposed method with different annotator's types from those in the main paper. Specifically, for simulated annotations for target tasks, we considered hammers, pair-wise flippers, and class-wise spammers (Tanno et al., 2019; Khetan et al., 2018). Pair-wise flippers return correct labels with probability $q$ $(0.5 < q \leq 0.8)$ and flip the label of each class to another label with probability $1 - q$ (the flipping target was chosen uniformly at random at each class). Class-wise spammers can be experts with accuracy rate $q = 1$ for some subset of classes and spammers for the others. For each target task, we randomly selected two of four classes as the classes for the spammers. For the support set in each target task, we randomly draw $R$ annotators from a predefined annotator's distribution $p(A)$, where $A$ takes hammer (H), pair-wise flipper (P), or class-wise spammer (C). Specifically, we considered four cases $(p(\mathrm{H}), p(\mathrm{P}), p(\mathrm{C})) = (1.0, 0.0, 0.0), (0.0, 1.0, 0.0), (0.0, 0.0, 1.0),$

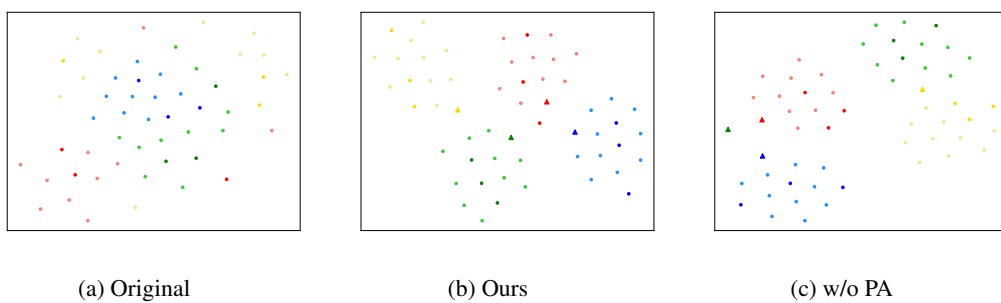

(a) Original                  (b) Ours                 (c) w/o PA

Figure 7: t-SNE (Van der Maaten & Hinton, 2008) visualization of embedded support and query examples obtained by the proposed method (Ours) and w/o PA in Omniglot. Each color represents a true class label. Darker and lighter colors represent support and query examples, respectively. Original represents original examples before performing embeddings. Although we visualized support data with truth class information, neither method uses these truth class information for training. The triangle represents the prototype for each class obtained from each method.

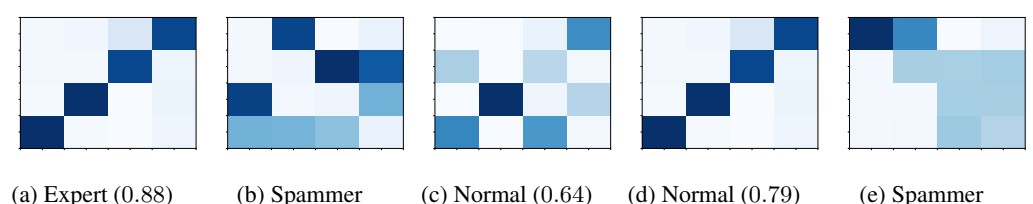

(a) Expert (0.88)     (b) Spammer     (c) Normal (0.64)     (d) Normal (0.79)     (e) Spammer

Figure 8: Estimated confusion matrices for annotators on a target task by the proposed method on Omniglot. We used four-class classification problem: three support examples per class and five annotators. Vertical and horizontal axes represent estimated and true class indexes, respectively. Each caption represents a true annotator type and a numerical value in the bracket represents accuracy rate $q$. Darker colors indicate higher values.

and $(0.33, 0.33, 0.33)$. After selecting the annotator type, we uniformly randomly selected the accuracy rate $q$ in the each range when the type was hammers or pair-wise flippers. All support data were annotated from all $R$ annotators. For the proposed method, pseudo-annotators were generated at each meta-training iteration from the annotator's distribution $(p(\mathrm{E}), p(\mathrm{H}), p(\mathrm{S})) = (0.1, 0.7, 0.2)$, where E, H, S represent experts, hammers, and spammers, respectively. Tables 3 and 4 show the average test accuracy with different numbers of support data and annotators on Omniglot and Mini-imagenet, respectively. The proposed method outperformed the other methods even when the target annotators' distributions were quite different from those in the meta-training phase.

## I.5 IMPACT OF THE DIRICHLET PARAMETER $b$

In the prior distributions of our model in Eq. (3), $(b, c, \tau)$ are treated as hyperparameters. As for $c$ and $\tau$, we set $c = 1$ and $\tau = 1$ for all experiments, which performed well. By tuning these parameters, the performance of the proposed method might be further improved. As for $b$, we evaluated the proposed method by changing $b$. Table 5 shows the average test accuracy on each dataset. We found that large $b$ slightly performed well. This is because $b$ controls the class prior of the support set. Specifically, as $b$ increases, the class priors $\boldsymbol{\pi} = (\pi_k)_{k=1}^{K}$ become uniform. Since the true class prior of the support set was uniform in our experiments, which is a standard setting in meta-learning studies (Snell et al., 2017; Finn et al., 2017; Rajeswaran et al., 2019; Bertinetto et al., 2018; Iwata & Kumagai, 2020), large $b$ worked well. We note that even small $b = 1$ outperformed other comparison methods (see Tables 1 and 2 in the main paper).

In addition, we explain that the proposed method can be robust against class-imbalance in target tasks when $b$ is sufficiently large. As explained before, the class prior $\boldsymbol{\pi} = (\pi_k)_{k=1}^{K}$ becomes uniform (i.e., $\pi_k = 1/K$) when $b$ is sufficiently large. Since $\pi_k$ represents the importance (weight)

Table 3: Average test accuracy with different numbers of support data and annotators on Omniglot with normal, pair-wise flippers, and class-wise spammers on target tasks. The number of classes in each task is four, and the number of support data per class (shot) was one, three, and five. Boldface denotes the best and comparable methods according to the paired t-test ($p = 0.05$).

| $N_S$ | $R$ | Ours | LRMV | LRDS | RFMV | RFDS | CL | CNAL | PrMV | PrDS | MaMV | MaDS | MCL | MCNAL | w/o PA |
|---|---|---|---|---|---|---|---|---|---|---|---|---|---|---|---|
| 4 | 3 | **0.788** | 0.449 | 0.460 | 0.368 | 0.380 | 0.630 | 0.465 | 0.752 | 0.779 | 0.742 | 0.769 | 0.734 | 0.567 | 0.467 |
| 4 | 5 | **0.893** | 0.488 | 0.487 | 0.408 | 0.410 | 0.684 | 0.523 | 0.851 | 0.847 | 0.839 | 0.835 | 0.825 | 0.665 | 0.502 |
| 4 | 7 | **0.923** | 0.511 | 0.514 | 0.432 | 0.434 | 0.731 | 0.525 | 0.904 | 0.908 | 0.890 | 0.894 | 0.877 | 0.676 | 0.529 |
| 12 | 3 | **0.945** | 0.544 | 0.572 | 0.472 | 0.508 | 0.758 | 0.697 | 0.893 | 0.901 | 0.773 | 0.796 | 0.840 | 0.798 | 0.840 |
| 12 | 5 | **0.982** | 0.613 | 0.629 | 0.542 | 0.560 | 0.847 | 0.767 | 0.952 | 0.960 | 0.856 | 0.881 | 0.925 | 0.877 | 0.923 |
| 12 | 7 | **0.990** | 0.657 | 0.669 | 0.582 | 0.600 | 0.884 | 0.792 | 0.979 | 0.984 | 0.913 | 0.926 | 0.957 | 0.905 | 0.957 |
| 20 | 3 | **0.971** | 0.594 | 0.628 | 0.546 | 0.588 | 0.814 | 0.765 | 0.938 | 0.946 | 0.793 | 0.836 | 0.881 | 0.843 | 0.949 |
| 20 | 5 | **0.989** | 0.667 | 0.702 | 0.617 | 0.656 | 0.893 | 0.830 | 0.978 | 0.985 | 0.883 | 0.916 | 0.949 | 0.914 | 0.983 |
| 20 | 7 | **0.993** | 0.718 | 0.746 | 0.666 | 0.696 | 0.928 | 0.857 | 0.991 | **0.995** | 0.934 | 0.957 | 0.975 | 0.940 | 0.990 |
| Avg. | | **0.942** | 0.582 | 0.601 | 0.515 | 0.537 | 0.797 | 0.691 | 0.916 | 0.923 | 0.847 | 0.868 | 0.885 | 0.798 | 0.793 |

Table 4: Average test accuracy with different numbers of support data and annotators on Miniimagenet with normal, pair-wise flippers, and class-wise spammers on target tasks. The number of classes in each task is four, and the number of support data per class (shot) was one, three, and five. Boldface denotes the best and comparable methods according to the paired t-test ($p = 0.05$).

| $N_S$ | $R$ | Ours | LRMV | LRDS | RFMV | RFDS | CL | CNAL | PrMV | PrDS | MaMV | MaDS | MCL | MCNAL | w/o PA |
|---|---|---|---|---|---|---|---|---|---|---|---|---|---|---|---|
| 4 | 3 | **0.413** | 0.243 | 0.248 | 0.256 | 0.260 | 0.297 | 0.271 | 0.394 | 0.405 | 0.388 | 0.392 | 0.346 | 0.304 | 0.329 |
| 4 | 5 | **0.460** | 0.241 | 0.245 | 0.263 | 0.263 | 0.295 | 0.284 | 0.429 | 0.427 | 0.414 | 0.312 | 0.364 | 0.317 | 0.354 |
| 4 | 7 | 0.454 | 0.243 | 0.243 | 0.264 | 0.264 | 0.308 | 0.285 | **0.455** | **0.458** | 0.431 | 0.432 | 0.385 | 0.314 | 0.390 |
| 12 | 3 | **0.576** | 0.291 | 0.296 | 0.277 | 0.283 | 0.373 | 0.342 | 0.473 | 0.499 | 0.452 | 0.460 | 0.452 | 0.401 | 0.416 |
| 12 | 5 | **0.606** | 0.301 | 0.303 | 0.285 | 0.290 | 0.389 | 0.356 | 0.534 | 0.554 | 0.496 | 0.504 | 0.488 | 0.436 | 0.468 |
| 12 | 7 | **0.647** | 0.309 | 0.310 | 0.298 | 0.301 | 0.412 | 0.367 | 0.580 | 0.593 | 0.536 | 0.542 | 0.522 | 0.450 | 0.514 |
| 20 | 3 | **0.640** | 0.318 | 0.324 | 0.316 | 0.330 | 0.418 | 0.382 | 0.532 | 0.568 | 0.477 | 0.501 | 0.516 | 0.464 | 0.525 |
| 20 | 5 | **0.662** | 0.331 | 0.336 | 0.334 | 0.340 | 0.451 | 0.403 | 0.598 | 0.624 | 0.533 | 0.559 | 0.562 | 0.510 | 0.591 |
| 20 | 7 | **0.697** | 0.339 | 0.346 | 0.343 | 0.355 | 0.470 | 0.413 | 0.631 | 0.657 | 0.570 | 0.598 | 0.593 | 0.519 | 0.632 |
| Avg. | | **0.573** | 0.291 | 0.295 | 0.293 | 0.298 | 0.379 | 0.346 | 0.514 | 0.532 | 0.477 | 0.489 | 0.470 | 0.413 | 0.469 |

Table 5: The proposed method with different Dirichlet parameters $b$. Average test accuracy over different numbers of support data and annotators on each dataset.

| Data | Ours ($b = 1$) | Ours ($b = 10$) | Ours ($b = 100$) |
|---|---|---|---|
| Omniglot | 0.884 | 0.890 | 0.892 |
| Miniimagenet | 0.534 | 0.540 | 0.542 |
| LabelMe | 0.503 | 0.512 | 0.520 |

of class $k$ during adaptation, $\pi_k = 1/K$ ensures that each class equally influences the adaptation, preventing the minority class from being ignored during adaptation (In contrast, since standard classification losses such as the cross-entropy loss minimize the average loss of all data, they can ignore the minority class's data). Overall, setting large $b$ would be preferable for the proposed method.

### I.6 COMPARISON WITH META-LEARNING METHODS WITH THE PSEUDO-ANNOTATION

In the main paper, we compared the proposed method with the meta-learning methods without the pseudo-annotation because there are no existing methods that use the pseudo annotation during meta-learning for learning multiple annotators. However, we also investigated comparisons with meta-learning methods that use the pseudo-annotation during meta-training phase. Table 6 shows the average test accuracy on each dataset. The symbol "w/ PA" indicates that the pseudo-annotation was used. MCL and MCNAL used the meta-learned embedding networks by the prototypical network with the pseudo-annotation. The proposed method outperformed the other methods on all datasets. These results suggest the effectiveness of our model design for learning how to learn from a few noisy labeled data obtained from multiple annotators.

### I.7 COMPARISON WITH META-LEARNING METHODS FOR LEARNING FROM NOISY LABELS

We compared the proposed method with a meta-learning method for learning from a few noisy data without modeling multiple annotators (Liang et al., 2022). Table 7 shows the average test accuracy on each dataset. LNL is the prototypical network-based meta-learning method and estimates class's prototypes by weighting each support example. The weight of each example is determined on the

Table 6: Average test accuracy over different numbers of support data and annotators on each dataset. Boldface denotes the best and comparable methods according to the paired t-test ($p = 0.05$).

| Data | Ours | PrMV w/ PA | PrDS w/ PA | MaMV w/ PA | MaDS w/ PA | MCL w/ PA | MCNAL w/ PA |
|---|---|---|---|---|---|---|---|
| Omniglot | **0.892** | 0.853 | 0.856 | 0.809 | 0.808 | 0.827 | 0.751 |
| Miniimagenet | **0.542** | 0.484 | 0.504 | 0.419 | 0.424 | 0.447 | 0.404 |
| LabelMe | **0.520** | 0.475 | 0.472 | 0.375 | 0.347 | 0.474 | 0.470 |

Table 7: Comparison with learning from noisy labels. Average test accuracy over different numbers of support data and annotators on each dataset. Boldface denotes the best and comparable methods according to the paired t-test ($p = 0.05$).

| Data | Ours | LNL (Liang et al., 2022) |
|---|---|---|
| Omniglot | 0.892 | **0.899** |
| Miniimagenet | **0.542** | 0.512 |
| LabelMe | **0.520** | 0.484 |

Table 8: Comparison with example-dependent confusion matrix-based methods. Average test accuracy over different numbers of support data and annotators on each dataset. Boldface denotes the best and comparable methods according to the paired t-test ($p = 0.05$).

| Data | Ours | LF (Gao et al., 2022) | MLF |
|---|---|---|---|
| Omniglot | **0.892** | 0.573 | 0.662 |
| Miniimagenet | **0.542** | 0.313 | 0.364 |
| LabelMe | **0.520** | 0.320 | 0.457 |

basis of the squared Euclidean distance between the example and other examples in the same (noisy) class to alleviate the harmful effect of the noisy examples. LNL uses the pseudo-annotation during the meta-learning phase like the proposed method. The proposed method outperformed LNL on Miniimagenet and LableMe by modeling multiple annotators. For Omniglot, the proposed method performed worse than LNL. Since Omniglot is relatively simple data, LNL may have worked well without modeling annotators.

## I.8 COMPARISON WITH EXAMPLE-DEPENDENT CONFUSION MATRIX-BASED METHODS

The proposed model assumes example-independent confusion matrix $p(y^r = l|t = k, \mathbf{A}) = \alpha_{lk}^r$, which is a standard assumption used in previous studies (Chu et al., 2021; Rodrigues & Pereira, 2018; Raykar et al., 2010; Tanno et al., 2019; Kim et al., 2022). However, some recent studies assume the example-dependent confusion matrix. Here, we compared the proposed method with Label Fusion (LF), a recent neural network-based method for multiple annotators using the example-dependent confusion matrix (Gao et al., 2022). We also created a meta-learning variant of LF (MLF), which used the pre-trained embedding networks with the prototypical networks for target tasks. For both methods (LF and MLF), we changed the number of basis matrices within $\{5, 10, 20\}$ and reported the best test results. Table 8 shows the average test accuracy on each dataset. The proposed method outperformed these methods by a large margin. This is because complex example-dependent confusion matrices are too difficult to estimate with limited data. This result suggests that our probabilistic modeling is well-suited for small data problems.

## I.9 EVALUATION ON CIFAR-10H

We additionally evaluated the proposed method on CIFAR-10H, which is a widely used real-world crowdsourcing dataset (Peterson et al., 2019). CIFAR-10H is image data, which consists of $10,000$ images of 10 classes. There are $2,571$ annotators and each annotator labels 200 images. Since such large-scale annotations are expensive and rare in practice, we selected a subset of low-quality annotators as in (Chu & Wang, 2021). Specifically, we selected 25 annotators from the annotators with the lowest annotation accuracy. Similar to the experiment with LabelMe, we used Miniimagenet for source tasks and CIFAR-10H for target tasks since CIFAR-10H does not have many classes. We

Table 9: Average test accuracies with different numbers of support data $N_S$ on CIFAR-10H. The number of classes in each task is ten, and the number of support data per class is one, three, and five. Boldface denotes the best and comparable methods according to the paired t-test ($p = 0.05$).

| $N_S$ | Ours | LRMV | LRDS | RFMV | RFDS | CL | CNAL | PrMV | PrDS | MaMV | MaDS | MCL | MCNAL | w/o PA |
|---|---|---|---|---|---|---|---|---|---|---|---|---|---|---|
| 10 | **0.147** | 0.128 | 0.129 | 0.117 | 0.117 | 0.137 | 0.137 | **0.150** | **0.148** | 0.131 | 0.129 | 0.139 | **0.146** | 0.111 |
| 30 | **0.184** | 0.147 | 0.144 | 0.139 | 0.137 | 0.160 | 0.160 | **0.183** | **0.182** | 0.152 | 0.153 | **0.175** | **0.176** | 0.127 |
| 50 | **0.223** | 0.154 | 0.150 | 0.148 | 0.142 | 0.171 | 0.173 | 0.207 | 0.204 | 0.158 | 0.159 | 0.205 | 0.207 | 0.143 |

Table 10: Results of the proposed method with a large number of support data. Average test accuracies over different annotator types when changing the number of target support data $N_S$ with $R = 7$ on Miniimagenet.

| $N_S$ | 20 | 40 | 80 | 120 | 160 |
|---|---|---|---|---|---|
| Our | 0.674 | 0.747 | 0.766 | 0.775 | 0.774 |

resized the images of CIFAR-10H to images with $84 \times 84$ pixels to match the image size. Table 9 shows the average accuracy on target tasks with different numbers of target support data with CIFAR-10H. The proposed method showed the best or comparable results in all cases.

## I.10 Results with Large Numbers of Support Data and Annotators

Although this paper focuses on learning from a few noisy data given multiple annotators, we investigated the performance of the proposed method with large numbers of support data $N_S$ and annotators $R$. In this experiment, we used Miniimagenet since it has many data per class (100). We set $N_S = 20$ and $R = 7$ during the meta-learning phase. First, we evaluated the proposed method by changing $N_S$ values with $R = 7$ in target tasks. Figure 10 shows the results. The performance increased as $N_S$ increased, and it seemed to converge when $N_S = 120$ or $N_S = 160$. Next, we evaluated the proposed method by changing $R$ values with $N_S = 40$ in target tasks. Figure 11 shows the results. Even if the number of support data is not so large ($N_S = 40$), the proposed method performed well with the large number of annotators $R$. This is because many annotators can improve label estimation performance.

## I.11 Detailed Computation Cost

Although the computation (meta-training and testing) time of the proposed method was already investigated in the main paper, we report more detailed results in Tables 12 and 13. Here, each method used the hyperparameter that was selected using validation data.

First, we compare the proposed method with non-meta-learning methods (LRMV, LRDS, RFMV, RFDS, CL, and CNAL) in Table 12. Although non-meta-learning methods do not require meta-learning computation, their accuracy is greatly inferior to ours. The testing times of all methods except CL and CNAL were fast. Since CL and CNAL are neural network-based methods, their testing times, which consist of both training time from support data and prediction time for test data in target tasks, were long. In summary, although the proposed method requires additional meta-learning time, it can significantly increase classification performance, and its adaptation and prediction on new target tasks are fast.

Next, in Table 13, we compare the proposed method with meta-learning methods (PrMV, PrDS, MaMV, MaDS, MCL, MCNAL, and w/o PA). MAML-based methods (MaMV and MaDS) took longer meta-learning time than the others. This is because MAML requires costly second-order derivatives of whole parameters of the neural network to solve the bi-level optimization as described in Section 2. Note that MAML is the most representative meta-learning method. Embedding-based methods (Ours, PrMV, PrDS, MCL, MCNAL, and w/o PA) had similar meta-learning times. Although our method's computational process is more complex than other embedding-based methods due to considering noisy annotators in the meta-learning phase, its meta-learning time is comparable to that of other efficient embedding methods because it performs fast adaptation with closed-form EM steps. Since PrMV, PrDS, MCL, MCNAL, and w/o PA (MaMV and MaDS) used the same meta-learning procedure, their meta-learning time are the same. After the meta-learning, the testing time of meta-learning methods, except for MCL and MCNAL, was fast. Since MCL and MCNAL

Table 11: Results of the proposed method with a large number of annotators. Average test accuracies over different annotator types when changing the number of annotators $R$ with $N_S = 40$ on Miniimagenet.

| $R$ | 7 | 10 | 20 | 30 |
|---|---|---|---|---|
| Our | 0.747 | 0.760 | 0.772 | 0.775 |

Table 12: Comparison of the proposed method and non-meta-learning methods on Omniglot dataset with $N_S = 4$ and $R = 5$. Testing time consists of training time from support data and prediction time for test data on target tasks.

| | Ours | LRMV | LRDS | RFMV | RFDS | CL | CNAL |
|---|---|---|---|---|---|---|---|
| Meta-train time [sec] | 1361.12 | - | - | - | - | - | - |
| Testing time [sec] | 0.960 | 0.679 | 0.734 | 3.995 | 4.520 | 112.19 | 166.21 |
| Accuracy | 0.814 | 0.456 | 0.458 | 0.382 | 0.384 | 0.641 | 0.483 |

Table 13: Comparison of the proposed method and meta-learning methods on Omniglot dataset with $N_S = 4$ and $R = 5$. Testing time consists of training time from support data and prediction time for test data on target tasks.

| | Ours | PrMV | PrDS | MaMV | MaDS | MCL | MCNAL | w/o PA |
|---|---|---|---|---|---|---|---|---|
| Meta-train time [sec] | 1361.12 | 1280.70 | 1280.70 | 3499.14 | 3499.14 | 1280.70 | 1280.70 | 1280.70 |
| Testing time [sec] | 0.960 | 0.928 | 0.947 | 2.185 | 2.192 | 113.59 | 166.25 | 0.948 |
| Accuracy | 0.814 | 0.769 | 0.775 | 0.758 | 0.764 | 0.754 | 0.593 | 0.458 |

required many fine-tuning iterations, their testing time was long. The accuracy of the proposed method is the best. In summary, the proposed method is superior in accuracy and efficiency.

### I.12 FULL RESULTS WITH STANDARD ERRORS

In the main paper, we only reported mean test accuracies in Tables 1 and 2 due to the lack of space. Here, we reported the full results including standard errors in Tables 14, 15, and 16. The proposed method outperformed the other methods. The standard errors of all methods tended to decrease as the number of support data or annotators increased since the large number of data and annotators generally leads to stable prediction.

## J  LIMITATIONS

The proposed method assumes that source tasks have clean labeled data as in almost all existing meta-learning methods (Snell et al., 2017; Finn et al., 2017; Garnelo et al., 2018; Rajeswaran et al., 2019; Bertinetto et al., 2018; Han et al., 2021a; Liang et al., 2022). Although this assumption might be restrictive in some cases, there are many applications where the proposed method can be applied as described in Section 1. When each source task has enough noisy labeled data, we can accurately estimate their ground truth labels by applying existing methods for learning from noisy labels (Han et al., 2018; 2020; Zheng et al., 2017). Therefore, in this case, we can use the source tasks with their estimated labels for the proposed method.

Although the proposed method worked well even when different datasets (e.g., Miniimagenet and LabelMe) or noise (annotator) models are used for source and target tasks in our experiments, when there is a significant discrepancy between the target and source tasks, meta-learning methods, including the proposed method, may not work well. This is a common challenge for existing meta-learning methods. To deal with this problem, using task-augmentation methods such as (Yao et al., 2021) in our framework would be one of the promising research directions.

Table 14: Average test accuracies and standard errors over four target annotator distributions with different numbers of support data $N_S$ and annotators $R$ on Omniglot and Miniimagenet. The number of classes is four, and the number of support data per class (shot) is one, three, and five.

| $N_S$ | $R$ | Ours | LRMV | LRDS | RFMV | RFDS | CL | CNAL |
|---|---|---|---|---|---|---|---|---|
| 4 | 3 | 0.692 (0.009) | 0.410 (0.004) | 0.422 (0.005) | 0.347 (0.004) | 0.358 (0.004) | 0.569 (0.007) | 0.425 (0.005) |
| 4 | 5 | 0.814 (0.007) | 0.456 (0.004) | 0.458 (0.004) | 0.382 (0.004) | 0.384 (0.004) | 0.641 (0.006) | 0.483 (0.005) |
| 4 | 7 | 0.855 (0.006) | 0.480 (0.004) | 0.485 (0.004) | 0.404 (0.004) | 0.410 (0.004) | 0.678 (0.006) | 0.485 (0.005) |
| 12 | 3 | 0.885 (0.007) | 0.498 (0.005) | 0.516 (0.005) | 0.438 (0.004) | 0.463 (0.004) | 0.700 (0.006) | 0.656 (0.006) |
| 12 | 5 | 0.938 (0.005) | 0.552 (0.005) | 0.568 (0.005) | 0.491 (0.004) | 0.511 (0.004) | 0.776 (0.005) | 0.719 (0.005) |
| 12 | 7 | 0.967 (0.003) | 0.608 (0.004) | 0.620 (0.004) | 0.539 (0.004) | 0.557 (0.004) | 0.836 (0.004) | 0.752 (0.005) |
| 20 | 3 | 0.930 (0.006) | 0.544 (0.005) | 0.562 (0.005) | 0.503 (0.005) | 0.535 (0.005) | 0.762 (0.006) | 0.732 (0.006) |
| 20 | 5 | 0.964 (0.003) | 0.606 (0.005) | 0.640 (0.005) | 0.566 (0.005) | 0.599 (0.005) | 0.837 (0.005) | 0.796 (0.005) |
| 20 | 7 | 0.982 (0.002) | 0.662 (0.004) | 0.688 (0.004) | 0.620 (0.004) | 0.644 (0.004) | 0.886 (0.004) | 0.826 (0.004) |

| $N_S$ | $R$ | PrMV | PrDS | MaMV | MaDS | MCL | MCNAL | w/o PA |
|---|---|---|---|---|---|---|---|---|
| 4 | 3 | 0.666 (0.009) | 0.687 (0.009) | 0.655 (0.008) | 0.678 (0.009) | 0.661 (0.008) | 0.511 (0.007) | 0.433 (0.006) |
| 4 | 5 | 0.769 (0.007) | 0.775 (0.007) | 0.758 (0.007) | 0.764 (0.007) | 0.754 (0.007) | 0.593 (0.007) | 0.458 (0.006) |
| 4 | 7 | 0.820 (0.007) | 0.834 (0.007) | 0.811 (0.007) | 0.823 (0.007) | 0.810 (0.006) | 0.606 (0.006) | 0.484 (0.006) |
| 12 | 3 | 0.825 (0.007) | 0.816 (0.008) | 0.698 (0.007) | 0.721 (0.007) | 0.778 (0.007) | 0.752 (0.006) | 0.794 (0.007) |
| 12 | 5 | 0.893 (0.006) | 0.891 (0.006) | 0.777 (0.006) | 0.799 (0.006) | 0.855 (0.005) | 0.819 (0.005) | 0.871 (0.006) |
| 12 | 7 | 0.943 (0.004) | 0.944 (0.004) | 0.847 (0.005) | 0.864 (0.005) | 0.912 (0.004) | 0.860 (0.005) | 0.924 (0.005) |
| 20 | 3 | 0.885 (0.006) | 0.871 (0.007) | 0.735 (0.007) | 0.760 (0.007) | 0.831 (0.006) | 0.805 (0.006) | 0.914 (0.006) |
| 20 | 5 | 0.936 (0.005) | 0.935 (0.005) | 0.817 (0.006) | 0.844 (0.006) | 0.900 (0.005) | 0.873 (0.005) | 0.959 (0.004) |
| 20 | 7 | 0.969 (0.003) | 0.971 (0.003) | 0.882 (0.005) | 0.900 (0.005) | 0.943 (0.003) | 0.903 (0.004) | 0.981 (0.002) |

(a) Omniglot

| $N_S$ | $R$ | Ours | LRMV | LRDS | RFMV | RFDS | CL | CNAL |
|---|---|---|---|---|---|---|---|---|
| 4 | 3 | 0.387 (0.005) | 0.245 (0.002) | 0.248 (0.002) | 0.256 (0.002) | 0.258 (0.002) | 0.287 (0.003) | 0.276 (0.002) |
| 4 | 5 | 0.436 (0.005) | 0.246 (0.002) | 0.245 (0.002) | 0.258 (0.002) | 0.259 (0.002) | 0.293 (0.003) | 0.282 (0.002) |
| 4 | 7 | 0.432 (0.005) | 0.243 (0.002) | 0.243 (0.002) | 0.262 (0.002) | 0.261 (0.002) | 0.301 (0.003) | 0.280 (0.002) |
| 12 | 3 | 0.534 (0.005) | 0.286 (0.003) | 0.286 (0.003) | 0.272 (0.002) | 0.277 (0.002) | 0.355 (0.003) | 0.331 (0.003) |
| 12 | 5 | 0.571 (0.005) | 0.297 (0.003) | 0.298 (0.003) | 0.285 (0.002) | 0.288 (0.002) | 0.381 (0.003) | 0.349 (0.003) |
| 12 | 7 | 0.621 (0.004) | 0.304 (0.003) | 0.304 (0.003) | 0.292 (0.002) | 0.294 (0.003) | 0.398 (0.003) | 0.356 (0.003) |
| 20 | 3 | 0.595 (0.005) | 0.311 (0.003) | 0.312 (0.003) | 0.304 (0.002) | 0.312 (0.003) | 0.397 (0.003) | 0.369 (0.003) |
| 20 | 5 | 0.628 (0.004) | 0.320 (0.003) | 0.327 (0.003) | 0.319 (0.002) | 0.329 (0.003) | 0.426 (0.003) | 0.392 (0.003) |
| 20 | 7 | 0.674 (0.004) | 0.332 (0.003) | 0.333 (0.003) | 0.336 (0.003) | 0.340 (0.003) | 0.448 (0.003) | 0.403 (0.003) |

| $N_S$ | $R$ | PrMV | PrDS | MaMV | MaDS | MCL | MCNAL | w/o PA |
|---|---|---|---|---|---|---|---|---|
| 4 | 3 | 0.374 (0.004) | 0.380 (0.004) | 0.365 (0.004) | 0.367 (0.004) | 0.331 (0.003) | 0.294 (0.003) | 0.316 (0.003) |
| 4 | 5 | 0.405 (0.005) | 0.405 (0.005) | 0.394 (0.004) | 0.392 (0.004) | 0.353 (0.004) | 0.308 (0.003) | 0.349 (0.004) |
| 4 | 7 | 0.432 (0.005) | 0.429 (0.005) | 0.409 (0.004) | 0.407 (0.004) | 0.369 (0.004) | 0.305 (0.003) | 0.372 (0.004) |
| 12 | 3 | 0.443 (0.005) | 0.464 (0.005) | 0.425 (0.005) | 0.428 (0.004) | 0.426 (0.004) | 0.390 (0.004) | 0.403 (0.004) |
| 12 | 5 | 0.494 (0.005) | 0.510 (0.005) | 0.457 (0.004) | 0.467 (0.004) | 0.466 (0.004) | 0.425 (0.004) | 0.442 (0.004) |
| 12 | 7 | 0.540 (0.005) | 0.556 (0.005) | 0.498 (0.004) | 0.506 (0.004) | 0.496 (0.004) | 0.434 (0.004) | 0.490 (0.005) |
| 20 | 3 | 0.485 (0.005) | 0.516 (0.005) | 0.437 (0.004) | 0.454 (0.005) | 0.490 (0.004) | 0.451 (0.004) | 0.500 (0.005) |
| 20 | 5 | 0.553 (0.005) | 0.579 (0.004) | 0.490 (0.004) | 0.512 (0.004) | 0.535 (0.004) | 0.495 (0.004) | 0.561 (0.005) |
| 20 | 7 | 0.600 (0.004) | 0.616 (0.004) | 0.536 (0.004) | 0.553 (0.004) | 0.564 (0.004) | 0.508 (0.004) | 0.606 (0.004) |

(b) Miniimagenet

Table 15: Average test accuracies and standard errors with different numbers of support data $N_S$ on LabelMe. The number of classes in each task is eight, and the number of support data per class is one, three, and five.

| $N_S$ | Ours | LRMV | LRDS | RFMV | RFDS | CL | CNAL |
|---|---|---|---|---|---|---|---|
| 8 | 0.414 (0.015) | 0.202 (0.012) | 0.208 (0.010) | 0.165 (0.008) | 0.173 (0.008) | 0.247 (0.011) | 0.240 (0.011) |
| 24 | 0.542 (0.011) | 0.261 (0.012) | 0.255 (0.010) | 0.243 (0.011) | 0.251 (0.014) | 0.359 (0.011) | 0.361 (0.011) |
| 40 | 0.605 (0.010) | 0.278 (0.011) | 0.271 (0.010) | 0.280 (0.010) | 0.276 (0.010) | 0.422 (0.012) | 0.426 (0.013) |

| $N_S$ | PrMV | PrDS | MaMV | MaDS | MCL | MCNAL | w/o PA |
|---|---|---|---|---|---|---|---|
| 8 | 0.381 (0.015) | 0.375 (0.016) | 0.297 (0.014) | 0.287 (0.015) | 0.329 (0.015) | 0.314 (0.014) | 0.276 (0.014) |
| 24 | 0.514 (0.012) | 0.508 (0.014) | 0.404 (0.015) | 0.411 (0.013) | 0.509 (0.012) | 0.488 (0.013) | 0.412 (0.018) |
| 40 | 0.576 (0.010) | 0.571 (0.011) | 0.460 (0.014) | 0.464 (0.015) | 0.592 (0.015) | 0.593 (0.012) | 0.515 (0.016) |

Table 16: Average test accuracies and standard errors with different numbers of support data $N_S$ on CIFAR-10H. The number of classes in each task is ten, and the number of support data per class is one, three, and five.

| $N_S$ | Ours | LRMV | LRDS | RFMV | RFDS | CL | CNAL |
|---|---|---|---|---|---|---|---|
| 10 | 0.147 (0.005) | 0.128 (0.004) | 0.129 (0.004) | 0.117 (0.003) | 0.117 (0.003) | 0.137 (0.004) | 0.137 (0.004) |
| 30 | 0.184 (0.004) | 0.147 (0.004) | 0.144 (0.004) | 0.139 (0.004) | 0.137 (0.003) | 0.160 (0.004) | 0.160 (0.005) |
| 50 | 0.223 (0.005) | 0.154 (0.004) | 0.150 (0.004) | 0.148 (0.003) | 0.142 (0.003) | 0.171 (0.004) | 0.173 (0.004) |
| $N_S$ | PrMV | PrDS | MaMV | MaDS | MCL | MCNAL | w/o PA |
| 10 | 0.150 (0.004) | 0.148 (0.004) | 0.131 (0.004) | 0.129 (0.004) | 0.139 (0.004) | 0.146 (0.004) | 0.111 (0.003) |
| 30 | 0.183 (0.005) | 0.182 (0.005) | 0.152 (0.004) | 0.153 (0.004) | 0.175 (0.005) | 0.176 (0.004) | 0.127 (0.004) |
| 50 | 0.207 (0.005) | 0.204 (0.004) | 0.158 (0.004) | 0.159 (0.004) | 0.205 (0.004) | 0.207 (0.005) | 0.143 (0.004) |

