# OpenReview forum: "Meta-learning Representations for Learning from Multiple Annotators"
_ICLR.cc/2025/Conference — Submitted to ICLR 2025_

### Official Review · Reviewer_yaqq · 2024-10-29

**Soundness:** 3
**Presentation:** 3
**Contribution:** 3
**Rating:** 6
**Confidence:** 4

**Summary:**

This paper proposes a meta-learning approach that utilizes noisy labels from multiple annotators to build a classifier without relying on true labels. The authors employ a probabilistic framework where latent class representations in a Mixture of Gaussians model are optimized via EM. This approach maximizes the likelihood of observed noisy labels given the latent variables, assuming these noisy annotations can guide the learning of true underlying classes.

**Strengths:**

1.I find this work’s approach novel in that it uses meta-learning based on a large amount of correctly labeled source data to effectively handle task-specific noisy data and learn about annotator errors.
2.The Bayesian model presented is well-structured and logically sound; adding a graphical model could further enhance clarity for readers.
3.I appreciate their attempt to encode annotator-specific classification offsets into prior knowledge, which, based on the experimental results, appears to be effective. This treatment convincingly demonstrates the model's potential to manage annotator noise while preserving classification accuracy.

**Weaknesses:**

1. The paper assumes isotropic variance in the latent space, simplifying computation but potentially limiting flexibility. Real-world data often exhibit complex, class-specific structures that may not align with uniform variance assumptions, particularly in nuanced classification tasks.
2. Modeling A as a K*K*r matrix may lead to over-parameterization, especially with limited data. Without visualization of learned matrices, it's challenging to assess whether A captures meaningful annotator patterns. Additionally, assuming A is feature-independent implies annotator errors are detached from data features, suggesting p(y|x) = p(y) . This could prevent the model from learning a classifier grounded in meaningful data representation.
3. Maximizing likelihood of noisy label data rather than true labels raises concerns. Likelihood maximization is traditionally aimed at learning distributions close to true labels; without this, it is unclear if maximizing the joint likelihood on noisy data allows meaningful inference of latent true labels. The model might be vulnerable to learning patterns in the noise rather than capturing true class structure.
4. The GMM-based approach assigns a Gaussian component per class, hindering efficiency in large class-count tasks. With high-dimensional data, EM’s complexity grows, making it computationally burdensome. Approaches like hierarchical GMMs or variational inference could enhance scalability without sacrificing interpretability.
5. The rationale and motivation behind the design of p(A) (e.g. why it utilizes the Dirichlet components) is not clearly demonstrated by the author.

**Questions:**

1.How does the choice of isotropic variance in the latent space affect the model's flexibility and accuracy? Would other types of covariance  provide a meaningful improvement in capturing complex data structures?
2.Can you show some of the learned Ar and demonstrate how effectively they capture the true annotation bias?
3.Can you further demonstrate why eq 3 is called a conjugate prior? I don't think the prior would have a closed analytical form as eq3. Then how can we say it is conjugate? Also, since we use an iterative method to approximate the posterior, does the conjugacy of the prior matter?

---

> ### Author Response · Authors · 2024-11-19
>
> Thank you for your positive comments and constructive feedback.
>
> > adding a graphical model could further enhance clarity for readers.
>
> Thank you for good suggestion. We have added the graphical model of our method in Section A of the revised paper.
>
> > The paper assumes isotropic variance in the latent space, simplifying computation but potentially limiting flexibility. Real-world data often exhibit complex, class-specific structures that may not align with uniform variance assumptions, particularly in nuanced classification tasks.
>
> > 1.How does the choice of isotropic variance in the latent space affect the model's flexibility and accuracy? Would other types of covariance provide a meaningful improvement in capturing complex data structures?
>
> Thank you for the insightful comment. As you mentioned, we used the isotropic variance in the latent space to simplify the modeling and its computation. However, since the latent space can be flexibly learned by a neural network, such simple modeling works well. Indeed, our method outperformed various comparison methods by a large margin, as shown in Tables 1 and 2.
> We believe that simple models with good performance are more preferable than complex models.
> Note that isotropic variance in the latent space is also assumed in prototypical networks, a well-known and high-performance embedding-based meta-learning method (Snell et al., 2017).
>
> Nevertheless, we can consider using a full covariance matrix for each class $\Sigma_k \in \mathbb{R}^{M \times M
> }$ on the latent space.
> Since the dimension of the latent space $M$ is arbitrarily determined by users, estimating the $M \times M$ matrix $\Sigma_k$ would not be a problem regarding computation costs when $M$ is not large. We have added the following sentence in Section 3.2 of the reviased paper.
>
> "Although we assume the isotropic variance ${\bf I}$ for simplicity, we can use other covariance matrices such as full covariance matrices."
>
> > Modeling A as a KKr matrix may lead to over-parameterization, especially with limited data. Without visualization of learned matrices, it's challenging to assess whether A captures meaningful annotator patterns.
> >  2.Can you show some of the learned Ar and demonstrate how effectively they capture the true annotation bias?
>
> We have already visualized the estimated annotator-specific confusion matrices in Figure 8.
> We found that the estimated confusion matrices can capture the characteristics of each annotator well.
> For example, in the estimated matrices of expert (a) and normal annotator with high labeling accuracy (d), the diagonal elements are large, meaning they tend to return correct labels. In contrast, the confusion matrices of annotators with low accuracy (b, c, e) have values in the off-diagonal ones, reflecting that these annotators assign noisy labels.
>
> > Additionally, assuming A is feature-independent implies annotator errors are detached from data features, suggesting p(y|x) = p(y) . This could prevent the model from learning a classifier grounded in meaningful data representation.
>
> We think that even when $A$ is input-independent (i.e., $p(y|t,x)=p(y|t)$), $p(y|x)$ is not equal to $p(y)$,
> where $x$ is data features, $y$ is its noisy label, and $t$ is its latent true label.
>
> We roughly explain this below.
> First, we have $p(y|x)= \sum_t p(y,t|x) = \sum_t p(y|t,x) p(t|x) = \sum_t p(y|t) p(t|x)$, where we used the assumption $p(y|t,x)=p(y|t)$ in the last equality.
> Next, we have $p(y)=\sum_t p(y,t) = \sum_t p(y|t) p(t)$.
> Thus, $p(y|x)-p(y) = \sum_t p(y|t) (p(t|x)-p(t))$.
> Since $p(t|x)$ is not equal to $p(t)$, we cannot conclude that $p(y|x)=p(y)$.
>
> > Maximizing likelihood of noisy label data rather than true labels raises concerns. Likelihood maximization is traditionally aimed at learning distributions close to true labels; without this, it is unclear if maximizing the joint likelihood on noisy data allows meaningful inference of latent true labels. The model might be vulnerable to learning patterns in the noise rather than capturing true class structure.
>
> Since we use the latent variable model in which the generation process of noisy labels from latent true labels is modeled (Eq. 1), maximizing the likelihood of noisy labeled data (more accurately, MAP estimation as described in Eq. 2) can lead to estimating true labels when modeling assumption (e.g., input-independent confusion matrices) is vaild.
> Indeed, this approach is commonly used for crowdsourcing studies and has shown excellent performance (Dawid
> & Skene, 1979; Chu et al., 2021; Rodrigues & Pereira, 2018; Raykar et al., 2010; Kim et al., 2022) as described in Section 3.
>
> Moreover, since our model is meta-learned such that it can improve the expected test performance after adaptation to noisy labeled data (i.e., inference of the latent variable model),
> our meta-learned model can infer more robustly against label noises by transferring knowledge on clean labeled data in source tasks.

---

> > ### Author Response · Authors · 2024-11-19
> >
> > > The GMM-based approach assigns a Gaussian component per class, hindering efficiency in large class-count tasks. With high-dimensional data, EM’s complexity grows, making it computationally burdensome.
> >
> > The time complexity of our EM algorithm is ${\mathcal O}(J N_{{\cal S}} K (KR + M))$, where $K$ is the number of classes, as described in Section 3.3.
> > Thus, when the number of classes $ K $ is large, our method may take a high computation cost, as you mentioned. However, in a typical few-shot learning setting, the number of classes is small (e.g., 5) (Bertinetto et al., 2018; Lee et al., 2019). In addition, since the dimension of latent space $M$ is controllable despite the input dimension of data $D$, it is also not a problem.
> >
> > [Lee et al, 2019] Lee, Kwonjoon, et al. "Meta-learning with differentiable convex optimization." CVPR. 2019.
> >
> > > Approaches like hierarchical GMMs or variational inference could enhance scalability without sacrificing interpretability.
> >
> > Thank you for good suggestion. We agree that using variational inference in our framework is a nice idea. We have added the following sentence in Section 5 of the revised paper.
> >
> > "We also plan to use variational Bayesian inference, which is also differentiable, instead of the EM algorithm in our framework."
> >
> > > 3.Can you further demonstrate why eq 3 is called a conjugate prior? I don't think the prior would have a closed analytical form as eq3. Then how can we say it is conjugate?
> >
> > Is the phrase 'prior would have...' in the above comment a typographical error for 'posterior would have...'? Assuming that is correct, We will respond below. If our interpretation is incorrect, please let me know.
> >
> > When the prior and the posterior distributions have the same form for a given likelihood function, the prior distribution is called a conjugate prior (Bishop, 2006).
> > By using conjugate priors, the inference of the probabilistic model can often be performed efficiently (Biship, 2006). Thus, conjugate priors are commonly used in probabilistic modeling.
> >
> > In our model, we called Eq. (3) conjugate prior distributions since each prior $p({\boldsymbol \mu}_k)$, $p({\bf A}_r)$, and $p({\boldsymbol \pi})$ is a conjugate prior for each component of the likelihood $p({\bf u} _n | t _n , {\bf M} )$, $p(y_n^r |  t_n, {\bf A})$, and $p(t_n | {\boldsymbol \pi})$, respectively.
> > Since our model has latent variable ${\bf t}$, the posterior distribution $p({\bf M}, {\bf A}, {\boldsymbol \pi}|{\bf U}, {\bf Y})$ would not have the same form as Eq. 3 as you commented. However, by using the conjugate priors, we can stabilize the EM algorithm, and obtain closed-form steps for the EM algorithm as described under Eq. 4.
> > To more accurately describe our prior distributions, we have revised the sentence above Eq. (3) as
> >
> > "where we assume that the conjugate prior distribution for each component of the likelihood:"
> >
> > > Also, since we use an iterative method to approximate the posterior, does the conjugacy of the prior matter?
> >
> > Yes. By using the conjugate priors (Eq. 3), the M step of the EM algorithm (Eq. 7) can be obtained as a closed-form.
> >
> > > The rationale and motivation behind the design of p(A) (e.g. why it utilizes the Dirichlet components) is not clearly demonstrated by the author.
> >
> > As described in the above answer, since the conjugate prior of the likelihood $p(y^r_n|t_n, A)$ (a multinominal distribution) is the Dirichlet distribution, we used $p(A_r)$ for the prior.
> > To clarify this, we have added the footnote in Section 3.2 of the revised paper.

---

> > > ### Comment · Reviewer_yaqq · 2024-11-26
> > > **Thank you for your rebuttal**
> > >
> > > Thank you for pointing out figure 8, I am sorry that I missed it in the first place.
> > > By questioning the feature-independent design of Matrix A, I intended to highlight that it is inappropriate to treat it as a model for explaining how annotators make mistakes. According to Matrix A, an observed label is determined solely by the transition probability p(y|t), where t is the true label, and it is completely independent of x. However, in practice, when annotators label data, they rely on something akin to p(y|x), as the features of the data (x) are their only basis for making labeling decisions. But I think this is minor issue and can be further improved in the future.
> > > Overall I think the paper is written technically well. In reality the pure source data might be difficult to find(and validate), But I think the proposed framework has the potential to relax the purity assumption and result in more practical applications. I would like to keep my positive rating on this paper.

---

> > > > ### Author Response · Authors · 2024-11-26
> > > >
> > > > Thank you for your response.
> > > >
> > > > We are pleased to hear that you acknowledge the quality of the technical writing and the practical potential of the proposed framework.
> > > >
> > > > We agree with your comment regarding the annotator's labeling mechanism.
> > > > In practice, annotators can consider not only the true label $t$ but also the data $x$ when assigning labels. But, since our paper explores a new problem setting (learning from a few noisy data given multiple annotators), we considered an input-independent confusion matrix $p(y∣t)$ as the first step. This simple modeling allowed us to easily solve the inner problem via the EM steps.
> > > > As you mentioned, we recognize incorporating input-dependent confusion matrices within our framework is an important challenge for the next step, as described in the conclusion.
> > > >
> > > > Thank you again for your thorough review of the paper. We believe your professional comments have improved the quality of the paper. If you have any further questions, please feel free to ask.

---

### Official Review · Reviewer_jTSk · 2024-11-04

**Soundness:** 3
**Presentation:** 2
**Contribution:** 2
**Rating:** 5
**Confidence:** 4

**Summary:**

The authors propose a meta-learning approach to improve classifier performance on tasks with limited, noisy annotations from multiple annotators. By embedding examples in a latent space and modeling class probabilities with a Gaussian Mixture Model (GMM), the method effectively leverages labeled data from source tasks to inform target tasks. The EM algorithm estimates annotator reliability, allowing the technique to backpropagate loss efficiently for meta-learning. The approach is tested on real-world datasets like LabelMe and synthetic datasets like Omniglot and Miniimagenet, showing robust improvements over competing methods.

**Strengths:**

* **Originality:** The paper takes an interesting approach to tackling noisy annotations in a way that considers both the variability in annotator reliability and the scarcity of target task data. By simulating noisy labels during meta-training, the method closely approximates real-world challenges.

* **Quality:** The model is theoretically grounded, with empirical testing on multiple datasets to support its claims. Using the EM algorithm within meta-learning allows the model to adapt efficiently while controlling computational complexity. The comprehensive experiments add credibility to the method’s robustness across noise levels.

* **Clarity:** Overall the paper is well-organized.

* **Significance:** Although the proposed method requires additional samples from the source task, it could benefit numerous fields that rely on crowdsourced or expert-annotated data.

**Weaknesses:**

* **Computational Complexity:** The meta-learning phase involves training over multiple source tasks and iterative EM updates and maybe computationally demanding on larger datasets or in high-dimensional settings.

* **Impractical Settings** Besides, such high-quality source samples in the source tasks might not be available in practice. Even in most existing test sets (i.e., CIFAR-10, CIFAR-100, ImageNet), there are a lot of label error issues as mentioned in R1. Noting that the testing phase still struggles to find high-quality annotated data, how can we assume perfect annotations are ready in so many source tasks?

* **Annotation Modeling Limitation:** By using input-independent confusion matrices, the model does not account for instance-specific annotator behavior, potentially missing nuances in how annotator quality varies by data complexity. Extending to instance-dependent matrices (R2, R3) could improve performance since real-world noise is observed to be more related to instance-dependent label noise (R3).

* **Clarity in Paper Presentation:** The paper presentation could be improved significantly by adding more clarifications. (Please refer to the "Questions")

**Reference:**

R1: Pervasive Label Errors in Test Sets Destabilize Machine Learning Benchmarks. Neurips 2021.

R2: Part-dependent label noise: Towards instance-dependent label noise. Neurips 2020.

R3: Learning with Instance-Dependent Label Noise: A Sample Sieve Approach. ICLR 2023.

R4: Learning with Noisy Labels Revisited: A Study Using Real-World Human Annotations. ICLR 2022.

**Questions:**

Q1: In Lines 14-15, the authors mentioned that:

``` To learn accurate classifiers, existing methods require many annotated data to deal with noisy labels.```

It could be better if additional revision is applied to this sentence, since the mentioned annotated data itself is often viewed as data with noisy labels, due to imperfect expertise.

Q2: The naming of the "testing phase" in Figure 1 is confusing.

The paper talks about learning from multiple noisy annotations, here "testing" refers to the testing of the meta-learned model or the testing of the model trained on adapted labels.

Q3: Clarification of the inner problem [Line 69]

Empirically, what are the task differences here? Ideally, if the tasks are disjoint, then the inner problem of the bi-level optimization results in pretty high (i.e., 1) weight to the same/similar task, and almost no weight for the rest.

Q4: Impractical assumptions

In Line 77, the authors mentioned that ```Since source tasks contain only clean labeled data,```

This seems to be a harsh assumption. Even in most existing test sets (i.e., CIFAR-10, CIFAR-100, ImageNet), there are a lot of label error issues as mentioned in R1.

Q5: Line 82: ```Gaussian misture model (GMM)``` $\rightarrow$ Gaussian Misture Model (GMM)

Q6: In Lines (103-104), the authors mentioned that

```However, it often does not work well because it ignores per-annotator characteristics (Raykar et al., 2010).```

Agree that label aggregation will ignore per-annotator characteristics and lose too much information. However, there is a list of solutions that consider soft labels instead of aggregated labels. It is better to include the discussion/experiment comparisons as well. (i.e., simply average the multiple labels by assuming same expertise among annotators)

Q7: In Lines 127-129, the authors mentioned that

```the proposed method can adapt to data given from noisy annotators via the closed-form EM steps, which leads to effective and efficient meta-learning.```

I partially disagrees with authors that the proposed method leads to "efficient" meta learning, cuz it requires high quality annotations from source tasks.

Q8: Typo in Line 159. ```n-the``` $\rightarrow$ n-th

Q9: In line 222, it would be more beneficial to include more explanations on the assumption of conjugate prior distributions.

Q10: In Equation 5, it would be more beneficial to include more discussions about the tightness (or the performance gap) of the inequation.

Q11: In Line 272, the input of the algorithm 1 requires the confusion matrix distribution, which is often assumed to be a quite hard condition.

Q12: In lines 382-383, the experiment result would benefit a lot if including instance-dependent noise or real-world noise as well, for the annotator's type.

---

> ### Author Response · Authors · 2024-11-19
>
> Thank you for your insightful and detailed comments and constructive feedback.
>
> > Computational Complexity: The meta-learning phase involves training over multiple source tasks and iterative EM updates and maybe computationally demanding on larger datasets or in high-dimensional settings.
>
> We agree that meta-learning methods, including our method, generally take more computation cost than ordinary machine-learning methods, especially when data in source tasks are large. However, meta-learning methods can improve the performance greatly as shown in Tables 1 and 2.
> In addition, the meta-learning time of our method is significantly shorter than existing gradient-based meta-learning methods (MAML) due to the efficient closed-form EM steps, which is demonstrated in Sections 4.3 and I.11. Although our method requires iterative EM updates, Figure 6 shows that our method even with small EM steps (2 or 3) performed best. Even if the input dimension is large, since our EM steps are performed on the $M$-dimensional latent space where users can control the dimension, the computation cost of our EM algorithm is low when $M$ is not large.
>
> > Impractical Settings Besides, such high-quality source samples in the source tasks might not be available in practice. Even in most existing test sets (i.e., CIFAR-10, CIFAR-100, ImageNet), there are a lot of label error issues as mentioned in R1. Noting that the testing phase still struggles to find high-quality annotated data, how can we assume perfect annotations are ready in so many source tasks?
>
> > Q4: Impractical assumptions
> >In Line 77, the authors mentioned that Since source tasks contain only clean labeled data,
> >This seems to be a harsh assumption. Even in most existing test sets (i.e., CIFAR-10, CIFAR-100, ImageNet), there are a lot of label error issues as mentioned in R1.
>
> This is an important comment. As you mentioned, label noise is inevitable in real-world data. Therefore, assuming completely clean labeled data may be overly simplistic. In the paper, for the sake of convenience, we treated the labels in commonly used datasets for classification tasks (e.g., Omniglot and MiniImagenet) as correct. This assumption is also commonly adopted in experiments of existing studies on meta-learning, crowdsourcing, and noisy label learning (Finn et al., 2017; Snell et al., 2017; Hospedales et al., 2020; Chu et al., 2021; Gao et al., 2022; Guo et al., 2023).
>
> While label noise is inherent in real-world data, the extent of noise can vary depending on the dataset. For instance, the mean labeling accuracy is 69.2% in the LabelMe dataset used as target tasks in our experiments (Chu & Wang, 2021), while it is approximately 96.7% in the standard datasets evaluated in [R1].
> Therefore, in practice, to improve performance on datasets with high noise rates such as LabelMe, it would be reasonable to use datasets with relatively low noise rates as the source tasks. Standard datasets can be assumed to have relatively low noise levels and are thus appropriate for use.
> To clarify this point, we have added the following note in Section 1 of the revised paper:
>
> "For convenience, throughout this paper, we assume that all labels in the source tasks are correct. However, it may be difficult to collect entirely clean data in real-world applications. Thus, in practice, we use datasets that are expected to have relatively low levels of noise as the source tasks."
>
> > Annotation Modeling Limitation: By using input-independent confusion matrices, the model does not account for instance-specific annotator behavior, potentially missing nuances in how annotator quality varies by data complexity. Extending to instance-dependent matrices (R2, R3) could improve performance since real-world noise is observed to be more related to instance-dependent label noise (R3).
>
> Thank you for the suggestion. This paper proposes a new problem setting (i.e., meta-learning for learning from a few noisy data given multiple annotators). As a first step in solving this problem, it is better to consider simple input-independent matrices rather than complex models. Even with this simple modeling, we have confirmed that our method outperforms methods that use input-dependent matrices (LF and MLF) in Table 8.
>
> Although the input-dependent confusion matrices could potentially improve performance as you mentioned, they might have drawbacks in terms of their efficiency in solving inner problems.
> Thus, as described in the conclusion, extending our method to handle input-dependent confusion matrices will challenge future work.

---

> > ### Author Response · Authors · 2024-11-19
> >
> > > Q1: In Lines 14-15, the authors mentioned that:
> > > "To learn accurate classifiers, existing methods require many annotated data to deal with noisy labels."
> > > It could be better if additional revision is applied to this sentence, since the mentioned annotated data itself is often viewed as data with noisy labels, due to imperfect expertise.
> >
> > Thank you for good suggestion. We have revised this sentense as follows:
> >
> > "To learn accurate classifiers from such data, existing methods require many noisy annotated data."
> >
> > > Q2: The naming of the "testing phase" in Figure 1 is confusing.
> > > The paper talks about learning from multiple noisy annotations, here "testing" refers to the testing of the meta-learned model or the testing of the model trained on adapted labels.
> >
> > Thank you for the constructive suggestion. We have changed the term "Testing phase" to "Testing phase of the meta-leared model." in Figure 1 of the revised paper.
> >
> > > Q3: Clarification of the inner problem [Line 69]
> > > Empirically, what are the task differences here? Ideally, if the tasks are disjoint, then the inner problem of the bi-level optimization results in pretty high (i.e., 1) weight to the same/similar task, and almost no weight for the rest.
> >
> > In typical few-shot or meta-learning contexts, a task refers to a single classification problem, and the task difference refers to having different class labels (combinations). The inner problem is defined within a task, and our method does not consider task weights. Our method treats each source task equally and uses the source tasks to meta-learn a task-shared neural network for embeddings.
> >
> > > Q6: In Lines (103-104), the authors mentioned that
> > "However, it often does not work well because it ignores per-annotator characteristics (Raykar et al., 2010)."
> > > Agree that label aggregation will ignore per-annotator characteristics and lose too much information. However, there is a list of solutions that consider soft labels instead of aggregated labels. It is better to include the discussion/experiment comparisons as well. (i.e., simply average the multiple labels by assuming same expertise among annotators)
> >
> > Thank you for good suggestion. The comparison methods with the symbol 'DS' (LRDS, RFDS, PrDS, and MaDS) are equivalent to the methods in your comment when the number of EM step $J$ is one. This is because the initial values of estimated soft label $\lambda_{nk}$ in the EM algorithm were set to soft labels in your comment, and they were directly used for classifier learning when $J$ is one. These initial values were also used for our method, as described in Section 3.3. In our experiments, since $J$ was varied within $\\{1,2,3,4,5,10\\}$ and the best test result was reported within them for all comparison methods (except for our method that used validation data to determine hyperparameters), the results (average test accuracies) of the methods with soft labels are equivalent to or less than the results shown in Tables 1 and 2. Obviously, our method outperformed them.
> >
> > We have added the following sentence in Section 4.2 of the revised paper to clarify that the DS methods include such soft labels.
> >
> > "In the DS methods, the initial value of responsibility $\lambda_{nk}$ in the EM algorithm was set to $\lambda_{nk}= \frac{1}{R} \sum_{r=1}^{R} \delta (y^r_n,k)$, where $R$ is the number of annotators in a task, as in the proposed method.
> > When the number of the EM step $J$ is one, the initial values (soft labels) are directly used for classifier learning."
> >
> > > Q7: In Lines 127-129, the authors mentioned that
> > > "the proposed method can adapt to data given from noisy annotators via the closed-form EM steps, which leads to effective and efficient meta-learning."
> > > I partially disagrees with authors that the proposed method leads to "efficient" meta learning, cuz it requires high quality annotations from source tasks.
> >
> > We used the term "efficient" to indicate that our method excels in terms of computational cost (compared to other (gradient-based) meta-learning methods).
> > To clarify this, we have revised this sentence as follows:
> >
> > "the proposed method can adapt to data given from noisy annotators via the closed-form EM steps, which leads to effective and fast meta-learning."
> >
> > In addition, we note that the requirement of clean labeled data for meta-learning is common to other meta-learning methods (Finn et al., 2017; Snell et al., 2017; Hospedales et al., 2020; Rajeswaran et al., 2019; Bertinetto et al., 2018) not specific to our method.
> >
> > > Q8: Typo in Line 159. n-the -> n-th
> >
> > We have corrected this in the revised paper.

---

> > > ### Author Response · Authors · 2024-11-19
> > >
> > > > Q9: In line 222, it would be more beneficial to include more explanations on the assumption of conjugate prior distributions.
> > >
> > > When the prior and posterior distributions have the same distribution form, such prior distribution is called a conjugate prior (Bishop, 2006). For example, when the likelihood function is a Gaussian distribution, a Gaussian distribution is the conjugate prior. When the likelihood function is a multinomial distribution, the Dirichlet distribution is the conjugate prior.
> > >
> > > By setting conjugate priors, inferences of the probabilistic model can often be computed efficiently.
> > > Thus, conjugate priors are commonly used in probabilistic modeling. In our method, the conjugate priors (Eq. 3) can be used to derive the M-step of the EM algorithm in a closed form.
> > > To clarify this, we have added the footnote in Section 3.2 of the revised paper.
> > >
> > > > Q10: In Equation 5, it would be more beneficial to include more discussions about the tightness (or the performance gap) of the inequation.
> > >
> > > It is known that the tightness of the inequation depends on how each responsibility (the estimated probability of the class of $n$-th example is $k$) $\lambda_{nk}$ can approximate true posterior $p(t _n = k | {\bf u} _n, {\bf Y}, {\bf M}, {\boldsymbol \pi}, {\bf A} )$ (Bishop, 2006).
> > > The inequation becomes tighter as $\lambda _{nk}$ approaches the true posterior, and the equality holds if and only if $\lambda _{nk}$ matches the true posterior. Our EM algorithm can obtain the true posterior as a closed form in each E step (Eq. 6).
> > > To clarify this, we have added the following sentence in Section 3.2 of the revised paper.
> > >
> > > "This inequality becomes tighter as $\lambda_{nk}$ approaches true posterior $p(t _n = k | {\bf u} _n, {\bf Y}, {\bf M}, {\boldsymbol \pi}, {\bf A} )$, and the equality holds if and only if $\lambda _{nk}$ matches the true posterior (Bishop, 2006)."
> > >
> > > > Q11: In Line 272, the input of the algorithm 1 requires the confusion matrix distribution, which is often assumed to be a quite hard condition.
> > >
> > > The confusion matrix distribution $p(B)$, which serves as the input for Algorithm 1, can be arbitrarily set by the user. In our experiments, we demonstrated that even using a simple distribution, where the proportions of experts, normal, and spammers are 10%, 70%, and 20%, respectively, our method can perform well under various noise conditions such as the presence of class-wise spammers, pair-wise flippers, etc. (Tables 1, 3, and 4). Furthermore, our method also performed well on real-world annotated datasets (LabelMe and CIFAR10H) in Tables 2 and 9.
> > > While the performance of our method could potentially be further improved by using a more suitable confusion matrix distribution, studying how to determine the optimal distribution is one future work.
> > >
> > > Regarding the final point, we have added the footnote in Section 4.1 of the revised paper.
> > >
> > > "Although this simple distribution worked well in our experiments, there may be other distributions that are more optimal. Determining a better distribution will be a future challenge."
> > >
> > > > Q12: In lines 382-383, the experiment result would benefit a lot if including instance-dependent noise or real-world noise as well, for the annotator's type.
> > >
> > > Thank you for your constructive suggestion. We have evaluated our method with datasets with real-world noise (and possibly instance-denpendent noise) (LabelMe and CIFAR10H) in Tables 2 and 9. Our method outperformed the other methods in such conditions.

---

> > > > ### Author Response · Authors · 2024-11-26
> > > >
> > > > Dear Reviewer  jTSk,
> > > >
> > > > Thank you once again for taking the time to review our paper and for your valuable feedback.
> > > >
> > > > As the deadline for submitting the revised paper is approaching, we would like to confirm if our responses have addressed your concerns. If you have any further comments or questions, we would greatly appreciate it.
> > > >
> > > > Best regards,
> > > >
> > > > Authors

---

> > > > > ### Author Response · Authors · 2024-12-01
> > > > >
> > > > > Dear Reviewer  jTSk,
> > > > >
> > > > > We apologize for the consecutive submissions. As the discussion phase draws to a close, we hope we have addressed your concerns. However, if any aspects remain unresolved, we would be grateful if you could let us know.
> > > > > We would appreciate it if you could take another look at the paper, taking our responses into consideration.
> > > > > Thank you for your valuable time and effort in reviewing our paper.
> > > > >
> > > > > Best regards,
> > > > >
> > > > > Authors

---

### Official Review · Reviewer_UCB3 · 2024-11-04

**Soundness:** 2
**Presentation:** 1
**Contribution:** 2
**Rating:** 3
**Confidence:** 5

**Summary:**

The paper presents a meta-learning method designed to learn from multiple noisy annotators, addressing a common challenge in supervised learning where labels can vary in quality due to differences in annotator skills and biases. The proposed method aims to enhance classifier accuracy even when labeled data is limited. It achieves this by utilizing labeled data from related tasks, embedding examples into a latent space via a neural network, and constructing a probabilistic model to learn task-specific classifiers while assessing annotators' abilities.  The neural network is meta-learned to optimize test classification performance with small annotated datasets, leveraging the Expectation-Maximization (EM) algorithm for efficient adaptation. The effectiveness of the method is demonstrated through experiments on real-world datasets, including those with synthetic noise and actual crowdsourced data.

**Strengths:**

Originality: The paper introduces a novel approach that integrates meta-learning with the handling of noisy annotations, representing an innovative solution to a well-recognized problem in machine learning.
Quality: The methodology is technically sound, employing a clear framework and the well-established EM algorithm for parameter estimation. The approach is backed by empirical results, showcasing its effectiveness across various datasets.
Significance: This work is significant as it addresses a pressing issue in many practical applications, particularly in crowdsourcing scenarios where high-quality labeled data is hard to obtain. The proposed method could improve the robustness of machine learning models in real-world situations.

**Weaknesses:**

1.The description of this paper is not clear enough. For example, the description of the method proposed in this paper in the last two paragraphs of INTRODUCTION is lengthy and lacks logic. The RELATED WORK, DATA, COMPARISON METHODS sections are full of nonsense and lack structure.
2.The experimental comparison method lacks the latest methods in the last two years.
3.The datasets used in the experiment are all in the field of images and lack representativeness.

**Questions:**

There is an obvious challenge with this approach: how do we select and identify the source task 2?
Beyond that, why don't we just use transfer learning?

---

> ### Author Response · Authors · 2024-11-19
>
> Thank you for your insightful comments and constructive feedback.
>
> > 1.The description of this paper is not clear enough. For example, the description of the method proposed in this paper in the last two paragraphs of INTRODUCTION is lengthy and lacks logic.
>
> Thank you for the constructive suggestion.
> We have made the structure/description of Section 1 more evident by splitting the second-to-last long paragraph of Section 1 into two paragraphs. Specifically, in Section 1 of the revised paper, the third-to-last paragraph mainly explains the general mechanism of meta-learning (i.e., it consists of the inner and outer problems). The second-to-last paragraph focuses on the inner problem of our method. The final paragraph explains the outer problem of our method. By clearly defining each paragraph's subject, we believe that clarity/logic has been improved. If you have any other suggestions, we would appreciate your feedback.
>
> > The RELATED WORK, DATA, COMPARISON METHODS sections are full of nonsense and lack structure.
>
> Thank you for your feedback. We would appreciate it if you could provide more specific details regarding the issues of "full of nonsense" and "lack of structure." We want to incorporate the reviewers' comments as much as possible to improve the paper.
>
> In our submitted paper, we made each section have a clear structure.
> For example, in the related work section, we created three different paragraphs to explain three main related topics, i.e., methods for ordinary crowdsourcing (learning from mulitple noisy annotators), meta-learning methods for crowdsourcing, and general meta-learning methods, and discussed the position of our work in each paragraph.
> In the data section, we described the overview of datasets, how we created multiple tasks from these datasets, and the process of adding noisy annotations to each task by separating paragraphs.
> Similarly, in the comparing methods section, we first provided an overview of all comparison methods, followed by two separate paragraphs explaining comparison methods for ordinary crowdsourcing and for meta-learning of crowdsourcing.
>
> As indicated by the generally positive feedback from the other three reviewers on presentation and clarity (e.g., Reviewrs g7k1 and jTSk said that 'Presentation: 3: good' and Reviewr jTSk said that 'Clarity: Overall the paper is well-organized'), we believe the current structure is fine. However, if there are specific concerns, we would appreciate your feedback.
>
> > 2.The experimental comparison method lacks the latest methods in the last two years.
>
> We have compared many recent state-of-the-art crowdsourcing or noisy label learning methods in our experiments.
> Specifically, we evaluated the crowd layer (CL) (AAAI2018), the common noise adaptation layers (CNAL) (AAAI2021), few-shot learning from noisy labels (LNL) (CVPR2022), and Label Fusion (LF) (ECCV2022). In addition, since they are not designed for few-shot learning except for LNL, we also created their meta-learning variants (MCL, MCNAL, and MLF) based on the study (Zhang et al., 2023) and evaluated them. We compared 16 comparison methods in total. Our method outperformed these methods.
>
> Although there are more recent crowdsourcing methods such as (Guo et al., 2023; Li et al., 2024) as you mentioned, they all require large amounts of data. Thus, these studies differ from our work, which focuses on a small data regime. Our work proposes and explores a new problem different from traditional crowdsourcing studies, so we believe the current comparison methods are sufficient.
>
> > 3.The datasets used in the experiment are all in the field of images and lack representativeness.
>
> We agree that adding data other than image data in experiments can enrich our paper. However, in current machine learning research, image data are the standard benchmark, and many existing studies on crowdsourcing, noisy label learning, and meta-learning, including studies from top ML conferences such as NeurIPS, ICLR, ICML, AISTATS, and AAAI (Snell et al., 2017; Rajeswaran et al., 2019; Li et al., 2020; Bertinetto et al., 2018; Guo et al., 2023; Gao et al., 2022; Kim et al., 2022; Zhang et al, 2021), use only image data. Thus, we followed this. In our experiments, we evaluated our method with various conditions/perspectives, such as different numbers of support data, different numbers of annotators, different noise ratios, and different types of annotators.
> Thus, our experimental results are sufficiently reliable.
> In the future, we want to evaluate our method with data other than the image domain.
>
> [Zhang et al, 2021] Zhang, Yivan, Gang Niu, and Masashi Sugiyama. "Learning noise transition matrix from only noisy labels via total variation regularization." ICML 2021.

---

> > ### Author Response · Authors · 2024-11-19
> >
> > > There is an obvious challenge with this approach: how do we select and identify the source task?
> >
> > The choice of the source tasks depends on the application. If the target task is an image classification problem, as mentioned in Section 1, datasets with labeled images such as ImageNet can be used as the source tasks. If the target task is a text classification problem, various labeled text data from the web (e.g., text classification datasets from Kaggle or OpenML) can be used as the source tasks.
> > In meta-learning, the target and source tasks don't need to have the same class labels, and the class labels of the source tasks can be arbitrary. Thus, collecting source data would be relatively straightforward.
> > It is important to note that the challenges associated with source task selection are not specific to our work but apply to general meta-learning and transfer learning methods.
> >
> > > Beyond that, why don't we just use transfer learning?
> >
> > Meta-learning is a type of fine-tuning method in transfer learning, particularly effective in very small data regimes (Hospedales et al., 2020).
> > We focused on meta-learning since we wanted to learn from a small amount of noisy data.
> > Although the standard fine-tuning and meta-learning methods pre-train models with clean source data like the comparison methods in our experiments (PrMV, PrDS, MaMV, MaDS, MCL, MCNAL, and w/o PA),
> > we showed that it is insufficient in our problem setting. Our method can improve the performance with pseudo-annotation during the meta-learning.
> >
> > Domain adaptation is a well-known transfer learning approach.
> > Since it simultaneously uses source and target data for training, it requires time-consuming training as new target tasks appear. In contrast, meta-learning methods do not because the meta-learned model with source data can perform rapid and efficient adaptation with only target data.
> >
> > We have added this discussion in Section B of the revised paper.

---

> > > ### Author Response · Authors · 2024-11-26
> > >
> > > Dear Reviewer UCB3,
> > >
> > > Thank you once again for taking the time to review our paper and for your valuable feedback.
> > >
> > > As the deadline for submitting the revised paper is approaching, we would like to confirm if our responses have addressed your concerns.
> > > If you have any further comments or questions, we would greatly appreciate it.
> > >
> > > Best regards,
> > >
> > > Authors

---

> > ### Comment · Reviewer_UCB3 · 2024-11-29
> >
> > Thanks for the clarification. I agree that this paper does explore a new approach to traditional crowdsourcing research. However, I am still skeptical about the usefulness of this method, which is only applicable to the small data regime.

---

> ### Author Response · Authors · 2024-11-29
>
> Thank you for your reply.
>
> As you pointed out, our method focuses on the small data regime. In the context of learning from multiple annotators, the small data regime has not received much attention; however, as mentioned in Section 1, we believe there are many potential applications. For example, in domains such as security, healthcare, and science, where data is scarce and annotators must have deep expertise, the issue of small data with a few annotators is crucial (by the way, as mentioned in Section 1, even experts exhibit differences in annotations  (Raykar et al., 2010; Mimori et al., 2021).)
> Thus, we think our work is useful in addressing this challenge.
>
> When many annotated data are available, as you know, there are countless existing studies. Thus, such cases can be handled by the conventional methods. As the no-free-lunch theorem suggests, there is no universal algorithm for all cases. we think that it is important for the machine learning community to work together to increase the available tools that can be appropriately used in the right contexts.

---

### Official Review · Reviewer_g7k1 · 2024-11-04

**Soundness:** 2
**Presentation:** 3
**Contribution:** 2
**Rating:** 6
**Confidence:** 4

**Summary:**

This paper introduces a meta-learning approach to handle noisy labels from multiple annotators. Learning from multiple source tasks generalizes to new target tasks that also have noisy annotations, making it robust to labeling inconsistencies in real-world crowdsourcing settings. This paper simulates annotator noise with different noise types in the meta-training phase by introducing pseudo-annotators. These synthetic annotators add controlled noise to clean labels, enabling the model to learn to handle label noise in a simulated environment.

**Strengths:**

1. Unlike other meta-learning approaches like MAML, which uses second-order gradients, the proposed model uses the EM algorithm to achieve closed-form updates, making it computationally more efficient.

2. By training on multiple source tasks, the model learns how to handle new unseen noisy target tasks.

3. The pseudo-annotation strategy is interesting, where noise is artificially added to clean source data to simulate the real-world setting of noisy annotators.

**Weaknesses:**

1. Assumption: This paper assumes noise comes from multiple annotators with different error patterns and biases and each annotator’s behavior is consistent within a task, but for real-life data scenarios, it is not accurate, each annotator is domain-specific.

2. Novelty: The paper combines well-established methods like GMMs and annotator-specific confusion matrices within a meta-learning framework to handle noisy labels, the novelty is somewhat limited, such as GMM-based clustering has already been widely used for probabilistic class assignments and label noise mitigation.

**Questions:**

1. Increasing $N_s$ and $R$ significantly improves the model performance. The authors could add experiments with larger $N_s$ and $R$values to assess at what point performance stabilizes or converges.

2. How does the model perform if certain classes have significantly fewer support data points than others? How robust is it to imbalances in the number of labeled examples per class?

3. Discuss the situations if we have hundreds or thousands of annotators and each of them only label a few instances?

**Details Of Ethics Concerns:**

The author have already discussed it in their paper.

---

> ### Author Response · Authors · 2024-11-19
>
> Thank you for your insightful comments and constructive feedback.
>
> > Assumption: This paper assumes noise comes from multiple annotators with different error patterns and biases and each annotator’s behavior is consistent within a task, but for real-life data scenarios, it is not accurate, each annotator is domain-specific.
>
> Thank you for the insightful comment. The assumption that each annotator's behavior is consistent within a task is commonly used for modeling in crowdsourcing and noisy label learning studies (Chu et al., 2021; Rodrigues
> & Pereira, 2018; Raykar et al., 2010; Tanno et al., 2019; Kim et al., 2022). While real-world annotators may exhibit more complex behavior, the effectiveness of our method has been validated on datasets with real-world annotations (LabelMe and CIFAR10H) in Tables 2 and 9.
> Since this paper proposes a new problem setting (i.e., meta-learning for learning from a few noisy data given multiple annotators), it is reasonable to consider simple models as the first step.
> To better model real-world annotators and enhance our method's performance further, we can consider assuming input-dependent annotators' confusion matrices, one of the future works described in Section 5.
>
> > Novelty: The paper combines well-established methods like GMMs and annotator-specific confusion matrices within a meta-learning framework to handle noisy labels, the novelty is somewhat limited, such as GMM-based clustering has already been widely used for probabilistic class assignments and label noise mitigation.
>
> As you mentioned, GMM-based clustering and confusion matrices have been used in various tasks. However, combining them within meta-learning is novel, which all the other reviewers acknowledge. In particular, the important point is that our method is not just a simple combination; rather, it is cleverly designed to mitigate the high computational cost of meta-learning, a major problem of meta-learning methods described in Section 2.
> Specifically, by appropriately modeling the inner problem with GMM and confusion matrices, our method can efficiently solve the inner problem with the closed-form EM steps, which leads efficiently to solve the outer problem (meta-learning). The efficiency of our meta-learning are demonstrated in the last paragraph fo Section 4.3 and Section I.11.
>
> > Increasing $N_S$ and $R$ significantly improves the model performance. The authors could add experiments with larger $N_S$ and $R$ values to assess at what point performance stabilizes or converges.
>
> Thank you for the good suggestion.
> We have additionally investigated the performance of our method with large numbers of support data $N_S$ and annotators $R$.
> In this experiment, we used Miniimagenet since it has many data per class (100).
> We set $N_S=20$ and $R=7$ during the meta-learning phase.
> First, we evaluated our method by changing $N_S$ values with $R=7$ in target tasks. The average test accuracies are as follows.
>
> | $N_S=20$ | $40$ | $80$ | $120$ | $160$ |
> |---|---|---|---|---|
> | 0.674 | 0.747 | 0.766 | 0.775 | 0.774 |
>
> Our method's performance increased as $N_S$ increased, and it seemed to converge when $N_S=120$ or $N_S=160$.
> Next, we evaluated our method by changing $R$ values with $N_S=40$ in target tasks. The average test accuracies are as follows.
>
> | $R=7$ | $10$ | $20$ | $30$ |
> |---|---|---|---|
> | 0.747 | 0.760 | 0.772 | 0.775 |
>
> Even if the number of support data is not so large ($N_S=40$), our method performed well with the large number of annotators $R$.
> This is because many annotators can improve label estimation performance.
> We have added these results in Section I.10 of the revised paper.
>
> > How does the model perform if certain classes have significantly fewer support data points than others? How robust is it to imbalances in the number of labeled examples per class?
>
> Our method can be robust against data imbalance in target tasks, as explained below.
>
> In our model, mixture ratio $\pi_k$ (to be estimated) represents the importance (weight) of class $k$ during adaptation for the inner problem.
> In Eq. (7) of the EM steps, by setting the Dirichlet parameter's (hyperparameter) $b>0$ sufficiently large values, we can get
>
> $\pi _k = \frac{ \sum _{n=1}^{N _{\cal S}} \lambda _{nk} + b }{Kb + N _{\cal S} } = \frac{ \sum _{n=1}^{N _{\cal S}} \frac{\lambda _{nk}}{b} + 1 }{K + \frac{N _{\cal S}}{b} } \approx \frac{1}{K}$,
>
> where $K$ is the number of classes.
> This ensures that each class equally influences the adaptation, preventing the minority class from being ignored during adaptation (In contrast, standard classification losses such as the cross-entropy loss can ignore the minority class's data since they minimize the average loss of all data.)
>
> We have added these discussion in Section I.5 of the revised paper.

---

> > ### Author Response · Authors · 2024-11-19
> >
> > > Discuss the situations if we have hundreds or thousands of annotators and each of them only label a few instances?
> >
> > We explain this situation in two cases.
> >
> > When each annotator labels 'the same' support data in a target task, accurate labels of the data can be estimated since each instance has many annotations. Thus, in such cases, our method and conventional meta-learning methods that simply use support data with the estimated labels in taget tasks (e.g., PrMV, PrDS, MaMV, and MaDS) would perform well. In contrast, since the number of support data is small, ordinary crowdsourcing methods that do not use source data cannot learn accurate classifiers even if they can estimate accurate labels of support data.
> >
> > When each annotator labels (generally) 'different' support data in a target task, the number of whole support data becomes large since there are many annotators. Since we focus on a small data regime, this situation is out of the scope of our work.
> > Ordinary crowdsourcing methods such as [a] are more appropriate for these situations.
> >
> > [a] Zhang, Hansong, et al. "Coupled confusion correction: Learning from crowds with sparse annotations.", AAAI2024.
> >
> > We have added these discussions in Section E of the revised paper.

---

> > > ### Author Response · Authors · 2024-11-26
> > >
> > > Dear Reviewer g7k1,
> > >
> > > Thank you once again for taking the time to review our paper and for your valuable feedback.
> > >
> > > As the deadline for submitting the revised paper is approaching, we would like to confirm if our responses have addressed your concerns.
> > > If you have any further comments or questions, we would greatly appreciate it.
> > >
> > > Best regards,
> > >
> > > Authors

---

> > > > ### Author Response · Authors · 2024-12-01
> > > >
> > > > Dear Reviewer g7k1,
> > > >
> > > > We apologize for the consecutive submissions. As the discussion phase draws to a close, we hope we have addressed your concerns. However, if any aspects remain unresolved, we would be grateful if you could let us know.
> > > > We would appreciate it if you could take another look at the paper, taking our responses into consideration.
> > > > Thank you for your valuable time and effort in reviewing our paper.
> > > >
> > > > Best regards,
> > > >
> > > > Authors

---

### Author Response · Authors · 2024-11-19

Dear reviewers,

Thank you very much for taking the time to read our paper and for providing valuable feedback. Based on your comments, we have revised the paper accordingly. The changes have been highlighted in blue for your convenience.

We greatly appreciate your time and consideration.

Best regards,

Authors

---

### Meta-Review · Area_Chair_K4KR · 2024-12-23

**Metareview:**

Summary
The paper proposes a meta-learning method for training classifiers using data annotated by multiple noisy annotators. Unlike traditional approaches that require large amounts of noisy data for effective learning, this method leverages labelled data from related tasks. It embeds examples into a latent space, constructs task-specific classifiers, and estimates annotator reliability. An EM algorithm with a closed-form solution, efficiently implemented via backpropagation, is used for optimization. By simulating various annotator noise types through pseudo-annotators during the meta-learning phase, the model gains robustness to similar label noise. The approach is validated on both synthetic and real-world crowdsourcing datasets.

Strengths
The paper's strengths include using EM in meta-learning, which improves efficiency, and showing strong performance on unseen noisy tasks. Reviewers have also mentioned the novelty and significance of the problem, recognizing its contribution to handling noisy annotations effectively.

Weakness
A key issue is the approach's strong assumptions, such as relying on clean source data and using a simplistic annotator model, which may limit its practical applicability. Reviewers noted that these assumptions could make the method less relevant in real-world noisy scenarios, yet the paper does not provide empirical evidence to assess or mitigate this concern. This lack of validation leaves questions about whether the method's efficiency and design truly overcome its limitations in practice.

Recommendation
The paper addresses a significant and novel problem recognized by most reviewers. However, the overall reception has been mixed, with ratings of two sixes, one five, and one three after extended discussions. The primary concern lies in the strength of several assumptions underlying the proposed approach, which may limit its practical value. Unfortunately, the paper does not provide empirical evidence to address or evaluate these concerns, leaving key questions unresolved. Given these limitations, the paper does not fully meet the standards of a high-quality and comprehensive contribution, which may hinder its acceptance.

**Additional Comments On Reviewer Discussion:**

Reviewer UCB3 expressed concerns about the paper's presentation, particularly in the introduction, related work, and experimental setting sections. The authors revised the paper to address these issues. However, UCB3 also questioned the inclusion of new methods and datasets in the experiments, suggesting additional comparison methods. The authors argued that these suggested methods require large amounts of data, which contrasts with their small-data focus, but they did not provide empirical evidence to substantiate this claim or show comparative results on the datasets. The reviewer found the lack of empirical studies, particularly ones demonstrating why the suggested methods fail on small data, unsatisfactory. This made the rebuttal unconvincing from UCB3’s perspective.

Reviewer g7k1 raised their score to 6 following the authors’ response, which included good answers and some empirical support.

Reviewer yaqq questioned the assumptions underlying the annotation ability matrix but maintained their score of 6 during discussions. While they were generally satisfied with the authors’ response, their concerns about the assumptions remained unresolved.

Reviewer jTSk gave the practical limitations of the method due to its reliance on clean source data and simplified annotator behavior. While acknowledging the novelty and significance of the work, jTSk maitained their score 5 after discussions. While it is reasonable to assume clean data for source tasks during method design, the paper should explore the method’s robustness under slightly violated assumptions. The absence of such exploration weakened their confidence in the practical utility of the approach.

Overall, the discussions revealed that the paper lacks solid empirical evidence regarding the scalability and robustness of the proposed method, particularly under more realistic conditions where some assumptions may not hold. This limitation, combined with concerns about presentation and practicality, led to the conclusion that the paper is not yet ready for publication as a high-quality contribution.

---

### Decision · Program_Chairs · 2025-01-22

Reject